# DISTRIBUTED PATE AND CAPC ON A DIET: PRIVATE KNOWLEDGE TRANSFER WITHOUT PUBLIC DATA OR PRIVATE INFERENCE

## ABSTRACT

The PATE algorithm is one of the canonical approaches to private machine learning. It leverages a private dataset to label a public dataset, enabling knowledge transfer from teachers to a student model under differential privacy (DP) guarantees. However, PATE's reliance on public data from the same distribution as the private data poses a fundamental limitation, particularly in domains such as healthcare and finance, where in-distribution public data is typically unavailable. In this work, we propose DIET-PATE which overcomes this limitation. Therefore, it combines programmatically generated data and data-free knowledge distillation. Our experiments demonstrate that DIET-PATE closely matches the performance of standard PATE, despite the absence of in-distribution public data. Furthermore, we show that our approach seamlessly extends to distributed collaborative learning with CaPC. In this setting, only PATE-based learning can be used to provide DP guarantees, as teacher models are trained by different entities and exchange knowledge solely via labels. By eliminating the need for in-distribution data during knowledge transfer, our method removes CaPC's reliance on private inference with encrypted data, substantially reducing computational overhead and, for the first time, enabling the use of more complex models and learning tasks. Moreover, leveraging programmatically generated data allows parties in CaPC to jointly train a global model, rather than just improving local ones, thereby achieving significantly higher utility. These advances extend the practicality of distributed private learning with PATE and CaPC to sensitive and complex domains.

## 1 INTRODUCTION

Privacy is crucial in machine learning since large amounts of data, including datasets from medical, financial, and other sensitive domains, are used to train models. The PATE algorithm (Papernot et al., 2017; 2018a) is one of the canonical algorithms to obtain privacy-preserving machine learning. PATE transfers the knowledge from a private teacher ensemble to a student model via a privacy-preserving labeling of public data. By limiting the impact of a given teacher on a final label and adding noise to the aggregated labels from the ensemble, PATE establishes rigorous $(\varepsilon, \delta)$-differential privacy guarantees (Dwork et al., 2006a).

However, its reliance on public data from the same distribution is a fundamental limitation of PATE that has often been criticized (Torkzadehmahani et al., 2019; Yoon et al., 2019). It has also prevented PATE's application in sensitive domains, such as medical or financial where such public data is usually unavailable. In this work, we propose a solution to overcome this limitation. To eliminate the dependence of PATE on public data, we introduce DIET-PATE for **D**ata-free **I**nformation **E**xtraction and **T**ransfer. Our method leverages two recent advances in data generation and knowledge transfer, which individually provide marginal gains, as we demonstrate in the empirical section. However, when combined, they deliver significant improvements through their synergy.

The first key ingredient in DIET-PATE is programmatically generated synthetic data. Such data is generated by using a collection of large-scale procedural image programs (Baradad et al., 2022). We use this data to pretrain the teacher models and the student, as well as to transfer the knowledge between them. The advantage of using programmatically generated data for pretraining is that it lets

the models learn generic features without consuming the privacy budget of the private data (Sander et al., 2024; Tang et al., 2023; Yu et al., 2024). This allows the full privacy budget to be used for teaching the models the specific features of the private data, maximizing its impact. Using the same programmatically generated data from pretraining also to transfer the knowledge to the student removes PATE's need of public data. However, this introduces a distribution shift as the teachers are trained on the private data.

To overcome the distribution shift, we rely on the second key ingredient in our DIET-PATE, namely a data-free knowledge distillation (DataFreeKD) to reduce the distribution shift. Raikwar & Mishra (2022) identified that the key limitation of distillation with data from a different distribution than the teacher's training set is the covariate shift in the distribution of hidden layer activations of the teacher model. They proposed to effectively reduce the covariate shift by using the current statistics instead of original statistics of the training data in the teacher's batch normalization layers. Applying DataFreeKD together with the programmatically generated data to PATE opens a new way to perform a private knowledge distillation without any need for public data from the same domain.

Our DIET-PATE seamlessly extends to a distributed setting, where each teacher is trained by a different party. Therefore, it can be integrated into the CaPC (Confidential and Private Collaborative) learning framework (Choquette-Choo et al., 2021), an extension of standard PATE. Note that in this distributed setting, only PATE and our DIET-PATE can be used to protect privacy, while the other canonical privacy framework for machine learning, namely DP-SGD (Abadi et al., 2016), is not applicable because the collaborating parties exchange only labels. More concretely, in CaPC, a distributed network of teacher models collaborates by exchanging label predictions in the PATE style, enabling each teacher to enhance its local model's performance while maintaining privacy. However, CaPC has two primary bottlenecks. The first one is the resource-intensive private inference (Boemer et al., 2020; Zhang et al., 2025), *i.e.,* teacher models having to perform inference on *encrypted* private samples. The second one is that the teachers improve their models separately, using private query data that cannot be shared publicly. Yet, each teacher consumes a fraction of the privacy budget. This causes modest improvements to several distributed models rather than a significant enhancement of a single model, limiting overall performance.

We demonstrate that our DIET-CaPC, the distributed version of DIET-PATE combined with CaPC, effectively eliminates these bottlenecks. First, by leveraging the synthetic data, we replace the costly private inference on encrypted data with the orders of magnitude faster standard inference. Second, since the query data now is synthetic—rather than private data from the individual parties as in CaPC—the queries and the labels from the ensemble can be publicly released. Using this data, a joint student model for all parties can be trained leveraging the entire privacy budget which significantly improves utility. Thus, our DIET-CaPC simultaneously improves efficiency and performance for all the collaborating parties.

In summary, we make the following contributions:

1. We introduce the DIET-PATE framework that eliminates the dependence of the canonical PATE framework on the availability of public data from the same distribution as the teacher models' private training data.

2. We show a synergy between the *programmatically generated data*, which we use to pretrain a student and teachers as well as transfer teachers' knowledge, and *data-free distillation* that aligns the activation distributions in private teacher ensemble during knowledge extraction.

3. We develop DIET-CaPC, which combines DIET-PATE and CaPC in the distributed setting. DIET-CaPC improves efficiency, by removing the need for costly private inference, and enables creation of a shared student model that simultaneously provides higher performance across all collaborating parties rather than only partially improving their local models.

4. We conduct extensive empirical evaluations, demonstrating the effectiveness of DIET-PATE and DIET-CaPC for central and distributed differentially private machine learning, respectively.

## 2 BACKGROUND AND RELATED WORK

**Differential Privacy (DP)** (Dwork et al., 2006a;b) is a mathematical framework that provides theoretical upper bounds on the privacy leakage that is incurred by running a randomized algorithm,

Figure 1: **Examples of programmatically generated data.** From left to right, StyleGAN-oriented, FractalDB, Dead Leaves mixed and Shaders21k Mixup, two examples per dataset are shown.

such as training a model, on private data. Intuitively, DP ensures that no individual's data point significantly impacts the outcome of a computation. Formally, the privacy parameters $\varepsilon$ and $\delta$ are used to specify the privacy guarantee.

**PATE.** In this work, we achieve DP by post-processing the outputs of an ensemble of models trained on private data and using the noisy argmax mechanism introduced by Dwork et al. (2014), following the approach of *Private Aggregation of Teacher Ensembles* (PATE) (Papernot et al., 2017). In the noisy argmax mechanism, when queried on unlabeled public data, the teacher ensemble performs a private voting, followed by adding noise to the histogram of vote counts and returning the noisy label with the most votes. As an outcome from the PATE, a public student model is trained on the public data points with their corresponding noisy labels returned by the teacher ensemble. To compute the privacy guarantees (i.e., a bound on $\varepsilon$), PATE leverages the privacy analysis based on RDP (Mironov, 2017) as introduced by (Papernot et al., 2018a). See Appendix C.1 for an extended background on PATE. The key limitation of PATE is its dependence on the availability of public samples from the same distribution as the data used to train the teacher ensemble. In this work, we focus on how to overcome this problem.

**Confidential and Private Collaboration (CaPC).** The PATE algorithm was subsequently extended to the CaPC (Confidential and Private Collaborative) learning framework (Choquette-Choo et al., 2021), where a distributed set of teachers exchange the model predictions in the PATE style to improve their own local models. In CaPC (in step 1a of the protocol shown in Figure 11), a given participant $Q$ (querying party) encrypts a new unlabeled private example and sends it to all other collaborating (answering) parties, which in this case act like teachers in PATE; subsequently each of the teachers runs private inference on the encrypted example $\hat{x}$. The private inference (Boemer et al., 2020) is the most expensive part of CaPC, where the cryptographic techniques such as homomorphic encryption (HE) and secure multi-party computation (MPC) are combined to perform forward pass through a neural network (Boemer et al., 2020). This approach limits CaPC to small models due to these methods' computational complexity. The encrypted logits from all the answering parties are then sent to the querying party (step 1b), following the aggregation within MPC into a histogram (step 1c), where we add noise (step 2) and then release the final outcome to the querying party (step 3). We refer to Appendix C.3 for an extended background on CaPC. Our DIET-CaPC method enables us to create collaborative learning with a *standard inference* instead of the orders of magnitude more expensive *private inference* and to train a shared student model instead of only partially improving the separate models in each collaborating party. Additionally, without the heavy cryptographic operations for private inference, our DIET-CaPC caters to much more diverse and larger teacher or student models, thus opening the PATE-based collaborative learning to more complex problems.

**Programmatically Generated Data.** Programmatically generated data are synthetic images created via procedural image programs (Baradad et al., 2022). The generated outputs are systematically varied by modifying attributes such as color, size and combinations of shapes. Crucially, these generation programs can produce vast numbers of images far exceeding what can be manually created or collected, making scalability a defining advantage of programmatically generated data. From the privacy perspective, since no real-world samples are used in the process, the data does not incur any privacy leakage. In Figure 1, we plot some example data points generated through the different methods used throughout the paper. More examples, including the ones downscaled to 28x28 pixel and gray-scale are presented in Figure 7

**Knowledge Distillation (KD).** Knowledge Distillation is the process of transferring knowledge from one model, called teacher, to another one, denoted as a student. This strategy, mainly utilized to compress a large teacher model (Bucila et al., 2006), originally uses the teacher's training data (Hinton et al., 2015), which can lead to privacy risks. Instead of using the original data, Raikwar & Mishra

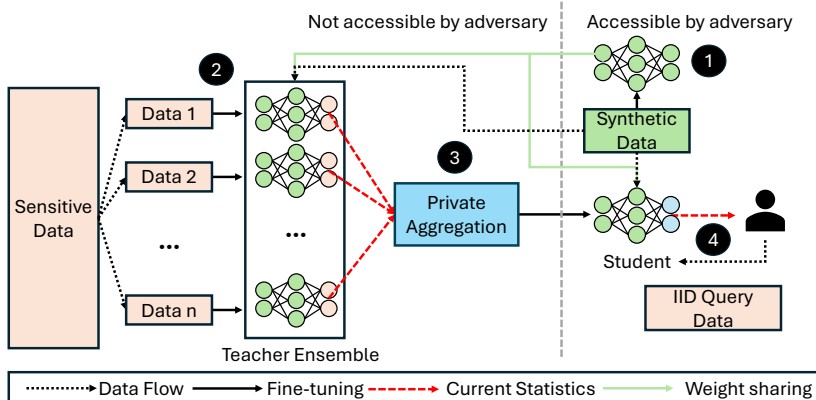

Figure 2: **Overview of DIET-PATE.** ❶ A model is pretrained on programmatically generated synthetic data. The model is distributed to all teachers and the student (green lines). ❷ Only teachers are fine-tuned on disjoint subsets of the sensitive data. ❸ A student model is trained on synthetic data via the data-free and private knowledge transfer via the aggregated teacher votes. This step utilizes the batch normalization layers in the teachers and the student to mitigate the covariate shift. ❹ The student can be queried with in-distribution data using the current statistics.

(2022) propose *data-free knowledge distillation* using Gaussian noise. Yet, utilizing Gaussian noise to transfer knowledge from one model to another poses challenges, due to the covariate shift in the inner activations of the neurons. Overall, the paper distinguishes between **1) current statistics** computed dynamically from the current mini-batch of data, *e.g.,* the Gaussian noise; and **2) original statistics** aggregated over the original training dataset. Raikwar & Mishra (2022) show that the covariate shift can be mitigated by adjusting the batch norm layers (Ioffe & Szegedy, 2015) and using the statistics of the current mini-batch during knowledge distillation.

## 3 OUR DIET-PATE FRAMEWORK

The DIET-PATE framework combines the strengths of the PATE approach with programmatically generated data and data-free distillation. Our method comprises four main stages, illustrated in Figure 2. These are:

❶ **Initialization:** In the first stage, the teacher models and the student model are initialized with the same weights. Therefore, we pretrain a single model on programmatically generated datasets, leveraging the diversity and scalability of this synthetic data. The pretrained model is then used as an initialization by all teachers and the student. In Section 5.3, we show that this joint initialization over teachers and student yields stronger final student model performance. This effect results from a closer prediction alignment in the behavior of teachers and student on the synthetic data.

❷ **Teacher Fine-Tuning:** In this stage, we partition the sensitive data into non-overlapping subsets, with each subset used to fine-tune the last layer of an individual teacher model.

❸ **Private Label Aggregation and Student Fine-Tuning:** In the third stage, we use synthetic samples and infer them one by one through the teacher ensemble. An important part of our framework is the decision on which statistics to use during training and inference to mitigate the covariate shift according to (Raikwar & Mishra, 2022). Each teacher does inference on a given programmatically generated sample using the **current statistics**, since the synthetic data is from a different distribution than the private data used to fine-tune the teachers. We show in Section 5.3 that the use of the **current statistics** is crucial to obtain a performant student model. The predictions from teachers are aggregated into a histogram. Following scalable PATE (Papernot et al., 2018a), we apply Confident-GNMax and return the noisy argmax, using Gaussian noise, as a label and then fine-tune the student on the synthetic samples and their respective noisy labels obtained from the teacher ensemble.

❹ **Student Model Deployment:** In the final stage, the fine-tuned student model is made publicly available. Users can query the student model with new data (IID Query Data) from the same distribution as the original sensitive teacher fine-tuning dataset. During these queries, the student model uses its **current statistics** to ensure accurate predictions. The **current statistics** should always be used during inference, when the training data and the query data are not IID.

The privacy guarantees in DIET-PATE follow directly from the ones in standard PATE (Papernot et al., 2018b), outlined in Appendix C.2. Following the original algorithm, we distribute the private data points into *disjoint* subsets for fine-tuning the teachers, limiting each data point's sensitivity in the same way as in the original algorithm. The use of programmatically generated data does not incur any additional privacy costs as this data is generated independently of any private data. In a similar vein, the teachers' current statistics are only modified based on the synthetically generated or the query data, which does not alter the aggregation rule and there is no overlap between teachers' private data. Finally, the student model only sees synthetic data paired with noisy aggregated labels that were obtained with privacy guarantees. Thus, equivalently as in the standard PATE algorithm, the student model training on the generated data pairs is a form of post-processing and, hence, does not incur additional privacy cost.

In summary, the core innovation of DIET-PATE lies in two key components, which individually provide marginal gains, as we demonstrate in the empirical section. However, when combined, they deliver significant improvements and strong privacy guarantees through their synergy.

## 4    Distributed Learning with Our DIET-CaPC

While the standard PATE framework assumes a centralized party that collects all the data, real-world scenarios often prohibit direct data sharing due to regulations, such as the GDPR (General Data Protection Regulation) European Parliament & Council of the European Union. To address this limitation, we propose DIET-CaPC, which combines DIET-PATE with the CaPC framework. We present the DIET-CaPC protocol and highlight how it overcomes the two major limitations of CaPC, making it truly applicable *in practice*.

### 4.1    CaPC and its Limitations

In CaPC each teacher corresponds to a different collaborating party, for example, different hospitals. Each hospital trains their own teacher model $i$ on locally obtained data. During collaboration, a given party $i$ can query the other collaborators acting as teacher ensemble to obtain labels for additional private data, allowing party $i$ to improve their local model. Unlike PATE, CaPC has no explicit student model, but every teacher acts as student model when it is their turn to query the ensemble.

To provide privacy for sensitive distributed applications, CaPC introduces additional protection measures that cause significant limitations: (1) CaPC uses the different parties' private data for querying the other parties. To prevent the other parties from accessing the private data, the data has to be encrypted. As a result, the other parties need to perform *resource-intensive private inference* on this encrypted data. (2) In CaPC, since the queries are private, they cannot be shared. Hence, each collaborating party can only use them to improve their own local model. Yet, each party's queries consume from the joint privacy budget. Hence, CaPC results in *only moderate improvements* to multiple distributed models instead of yielding a very strong shared model.

### 4.2    DIET-CaPC

DIET-CaPC combines the strengths of both, DIET-PATE and CaPC through multiple steps: (1) Each collaborating party's teacher model is initialized with the same pretrained model weights obtained from pre-training on the programmatically generated data. (2) The private query data from the different parties is replaced by programmatically generated query data. (3) Since this programmatically generated data has no privacy concerns and the teacher ensemble outputs a privatized label, the queries and final labels can be publicly released. This enables the training of a joint and public student model, just like in PATE. We describe the threat model of DIET-CaPC in Appendix D.

We distinguish between three types of parties in DIET-CaPC: (1) *Teacher Parties* are all the collaborators that want to jointly improve their predictions by training a joint student model. The models

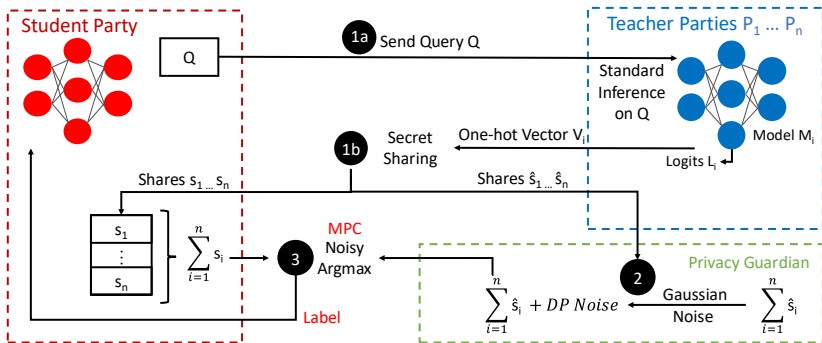

Figure 3: **DIET-CaPC Protocol.** **1a** A programmatically generated new sample $Q$ is sent to all answering parties $\mathcal{P}_i$, where $i \in [n]$. Each $\mathcal{P}_i$ runs standard inference on its model $\mathcal{M}_i$ and outputs (unencrypted, *i.e.,* plain) logits $\boldsymbol{L}_i$. We compute the one-hot vector $\boldsymbol{V}_i$ of the logits $\boldsymbol{L}_i$. **1b** Each answering party $\mathcal{P}_i$ generates secret shares of the one-hot vectors $\boldsymbol{V}_i$, distributing $\boldsymbol{s}_i$ to the student party and $\hat{\boldsymbol{s}}_i$ to the privacy guardian. **2** The privacy guardian sums $\hat{\boldsymbol{s}}_i$ from each $\mathcal{P}_i$ and adds Laplacian or Gaussian noise for DP. The student party sums $\boldsymbol{s}_i$ from each teacher party $\mathcal{P}_i$. **3** The privacy guardian and the student party run Yao's garbled circuit to obtain the argmax of the added shares from the student party and the noisy shares from the privacy guardian. Note that the added one-hot vectors $\boldsymbol{V}_i$ form a histogram. The label is returned to the student party.

trained by each party correspond to the teacher models in our DIET-PATE framework. (2) The ***Student Party*** can be one of the collaborators or any external party. The student party is responsible for engaging in the computation of the final label and for training the student model, which corresponds to the student model in DIET-PATE. (3) The ***Privacy Guardian***, just like in CaPC, is a third semi-trusted party that is responsible for protecting privacy of the outputs from the collaborating teacher parties (by adding Gaussian or Laplacian noise). This privacy guardian is needed as, in contrast to in DIET-PATE where all teachers and the student model are trained by the same party, in DIET-CaPC, different parties that cannot share their private data collaboratively train the models. We present DIET-CaPC's full distributed learning protocol in Figure 12 and Appendix D, and show the core protocol for DIET-CaPC in Figure 3. In the following, we detail how our DIET-CaPC overcomes CaPC's limitations.

**Synthetic Data Overcomes the Need for Private Inference.** DIET-CaPC solves the main issue in standard CaPC, which is the costly private inference on encrypted data, by leveraging the programmatically generated synthetic data. Since this synthetic data is non-private, it does not have to be encrypted. To train the student model, we simply generate a new synthetic data sample, and then **all teachers perform standard inference directly on this unencrypted sample**, which is presented to all the teachers in a plain form. Subsequently, DIET-CaPC follows the CaPC protocol and performs the private aggregation of answers from the collaborating (answering) parties, which guarantees the same privacy and security guarantees as in CaPC.

**A Single Shared Student Model Yields Higher Performance.** The newly labeled samples can be used to train a *shared* student model. The privacy budget in DIET-CaPC for all the private data is *jointly spent* across the teachers to label the newly generated synthetic data. In contrast, in CaPC, the privacy budget is *divided* among the teachers to answer their individual queries. This division of the privacy budget in CaPC is inefficient as it serves to marginally improve many local models instead of significantly improving a single model. Contrary, in DIET-CaPC, the whole ensemble of teachers consumes the full budget of $\varepsilon$ and transfers the entire resulting knowledge to the student model using all queries, hence benefiting from the main advantage of the full ensemble to obtain a higher performance than in any teacher. The shared student model can then also be used to label new locally obtained data and improve teacher $i$, without incurring any additional privacy cost. Thus, in all cases, our DIET-CaPC provides more benefits to the whole collaboration. DIET-CaPC becomes especially appealing in collaborative domains, where no party can share raw data but all benefit from a shared model.

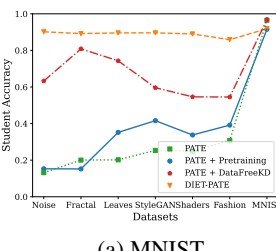 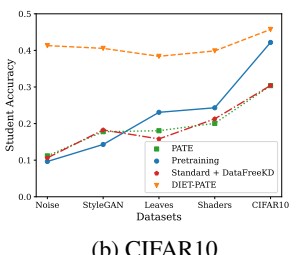 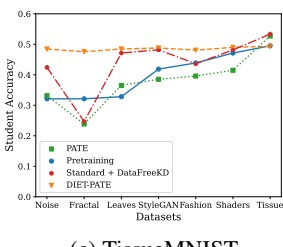

| (a) MNIST | (b) CIFAR10 | (c) TissueMNIST |

Figure 4: **DIET-PATE outperforms or matches the performance of PATE without using public data.** This figure shows the knowledge transfer capabilities of the different datasets with a privacy budget of $\varepsilon = 6$, $\delta = 10^{-5}$ for MNIST and $\varepsilon = 10$, $\delta = 10^{-5}$ for CIFAR10 and TissueMNIST.

## 5 EMPIRICAL EVALUATION

**Experimental Setup.** We evaluate DIET-PATE using the ResNet18 architecture (He et al., 2015) for all teachers and the student. We also run ablation studies where we follow the standard PATE and CaPC frameworks and use the ResNet10 architecture. The evaluation of DIET-CaPC is performed using the ResNet18 architecture for all teacher parties and the student party. We compare against two CaPC baselines specified in Section 5.4. The synthetic datasets considered in the experiments are Dead leaves mixed (Baradad et al., 2021), StyleGAN-Oriented (Baradad et al., 2021), FractalDB (Kataoka et al., 2021) and Shaders 21k MixUp (Baradad et al., 2022). We use MNIST (LeCun, 1998), CIFAR10 (Krizhevsky et al., 2009), and TissueMNIST (a collection of standardized biomedical images from Kidney Cortex Microscope with 236386 samples) (Yang et al., 2023) as private datasets. A full setup description for our experiments is included in Appendix A. We use the code from Raikwar & Mishra (2022) as the base for the student training while the training of teachers follows the standard PATE setup, including their privacy accounting (Papernot et al., 2018a).

### 5.1 DIET-PATE SIGNIFICANTLY IMPROVES PERFORMANCE

**Main Results.** DIET-PATE consistently outperforms or matches the performance of standard PATE without requiring public data. The main results, presented in Figure 4, show the accuracy of the student model when using programmatically generated or OOD data for knowledge transfer. On MNIST with $\varepsilon = 6$, DIET-PATE beats standard PATE by almost $7\times$ performance, with $90.181\%$ compared to $13.176\%$. Successful knowledge transfer is now possible even in the worst case setting when using only random noise, completely removing the necessity public data. Only in the setting with a sufficient amount of private data for training the teachers from scratch and sufficient same-distribution public data for the knowledge transfer (*e.g.,* red curve, right most data point in Figure 4a), standard PATE can outperform DIET-PATE. This is because PATE's teachers are trained directly on the in-distribution data (MNIST), allowing them to learn MNIST-specific features. In contrast, DIET-PATE 's teachers are pretrained on synthetic data, so they learn these features less precisely—a gap that last-layer fine-tuning cannot fully close. However in general, looking at all other datasets, we see that DIET-PATE achieves a constantly high performance and reliably outperforms standard PATE in the absence of public data.

**DIET-PATE under Different Privacy Regimes.** We also present the main results of our method for different $\varepsilon$ values in Figure 5. We see a consistent trade-off between privacy budget spent and accuracy gained, similar to standard PATE. While PATE achieves a better trade-off when public data from the same distribution is available (purple line), it completely fails in the absence of it (green line). In the case of CIFAR10, DIET-PATE even outperforms standard PATE on in-distribution public data. Still, for simpler tasks such as MNIST and TissueMNIST, a sufficient number of publicly available samples from the same distribution as the private data can boost PATE's performance, slightly exceeding the effectiveness of programmatically generated data in DIET-PATE. Yet, we observe that over all privacy regimes, DIET-PATE achieves a significantly better privacy-utility trade-off than PATE when no in-distribution public data is available, by leveraging the programmatically generated data.

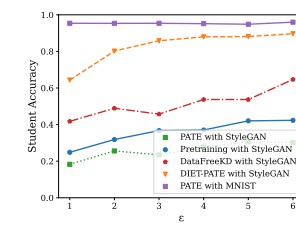 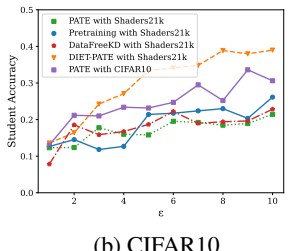 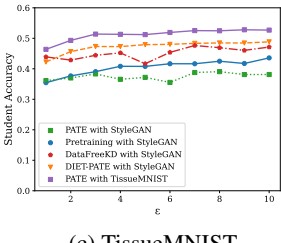

|              (a) MNIST              |             (b) CIFAR10             |           (c) TissueMNIST           |

Figure 5: **DIET-PATE exhibits much better privacy-utility trade-off than PATE when using only the programmatically generated data.** We present how the increased privacy budget corresponds to a higher accuracy of the student model.

## 5.2 EVALUATING DIET-PATE ON COMPLEX SETUPS

We also evaluate DIET-PATE on more challenging setups with a larger model and more complex private training data. Therefore, we instantiate both teachers and student with a vision transformer, TinyViT (Wu et al., 2022) (20.6 million parameters). We evaluate DIET-PATE on (1) a histopathologic cancer dataset (Cukierski, 2018), which consists of 220k images of size 96×96×3 and (2) the CheXpert dataset (Irvin et al., 2019), which consists of 224,316 images of size 320×320×3. For both datasets we use 80% of the data for teacher training. For CheXpert we perform a 3-class (negative, positive, unclear) classification for the *Lung Opacity* label, since DIET-PATE is not designed for multi-label classification. For both datasets we train 50 teachers for PATE and 50 teachers for DIET-PATE, where the latter were pretrained using Shaders21k. The full hyperparameter setup is provided in Appendix A.3. Our results, obtained over three random seeds, in Table 1 show that our DIET-PATE is effective on larger models and complex medical datasets. We observe that DIET-PATE consistently outperforms standard PATE by up to 30% (Gaussian noise, histopathologic cancer), highlighting our approaches ability to scale both in terms of data and model complexity.

Table 1: **DIET-PATE generalizes to larger datasets and model architectures.** We report the student validation accuracy in the histopathologic cancer dataset for $\varepsilon = 10, \delta = 5 \times 10^{-6}$.

|            | Histopathologic Cancer |                  |                  | CheXpert         |                  |                   |
| ---------- | ---------------------- | ---------------- | ---------------- | ---------------- | ---------------- | ----------------- |
| Method     | Gaussian Noise         | Shaders21k       | Public data      | Gaussian Noise   | Shaders21k       | Public data       |
| DIET-PATE  | $81.68 \pm 0.39$       | $81.37 \pm 0.53$ | $82.54 \pm 0.18$ | $68.59 \pm 2.46$ | $68.71 \pm 0.87$ | $70.844 \pm 0.69$ |
| PATE       | $49.48 \pm 5.04$       | $64.97 \pm 1.26$ | $76.40 \pm 1.10$ | $44.33 \pm 2.95$ | $50.55 \pm 3.30$ | $48.99 \pm 3.61$  |

## 5.3 ABLATION OF DESIGN CHOICES FOR THE SYNERGY IN DIET-PATE

**Joint Initialization of Teachers and Students Boosts Performance.** The first crucial part of DIET-PATE is using the same initialization for all teachers and the student. Table 2 depicts the accuracies of a ResNet10 student model when trained for MNIST with Gaussian noise with ($\varepsilon = 10, \delta = 10^{-5}$)-DP on labels returned by teachers with same or different initializations and the use of the current statistics.

Table 2: **Accuracy for same and different teacher (T) and student (S) model initializations.** We use PATE + DataFreeKD on ResNet10, for MNIST and transfer with Gaussian noise.

|                     | **T**:Same | **T**:Different |
| ------------------- | ---------- | --------------- |
| **S**:Same          | **63.6%**  | 51.5%           |
| **S**:Different     | 59.2%      | 51.4%           |

Our results highlight that with the same teacher and student initialization, DIET-PATE's performance improves. This is attributed to two reasons. (1) The same weight initialization in teachers causes higher consensus (Figure 9) between them. In PATE, this reduces the per-query privacy costs and yields more total answered queries. (2) The same initialization between teachers and students facilitate the knowledge transfer in the absence of public data.

**Finetuning only the Last Layer is Crucial.** To extend the previous ablation, we also ablate the choice of fine-tuning only the last layer of the student model and the teacher models. Therefore, we vary the fraction of parameters during fine-tuning, always starting from the last layer on the ResNet18

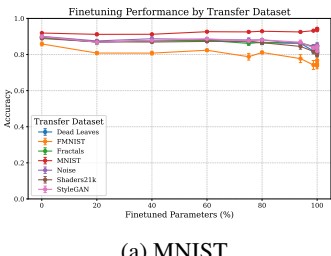
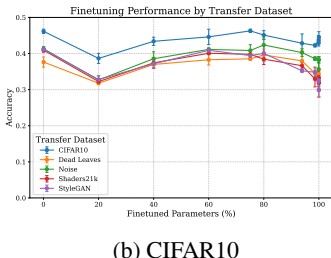

(a) MNIST                    (b) CIFAR10

Figure 6: **Finetuning only the last layer of the student provides overall the best performance.** We analyze the accuracy of the student when finetuning different percentage of the last parameters of the student model.

Table 3: **Finetuning only the last layer of student and teachers leads to the best performance with DIET-PATE.** We analyze the student model validation accuracy across five random seeds for the CIFAR10 dataset with ResNet18, when using Gaussian noise for the knowledge transfer.

| Finetuned Parameters (%) | 0.5 | 20 | 40 | 60 | 80 | 100 |
|---|---|---|---|---|---|---|
| Student fine-tuning | **43.32 ± 0.83** | 32.58 ± 0.66 | 39.61 ± 1.81 | 39.43 ± 0.77 | 40.54 ± 1.55 | 35.04 ± 2.31 |
| Student and Teacher fine-tuning | **43.32 ± 0.83** | 25.58 ± 1.37 | 32.45 ± 0.37 | 35.93 ± 2.83 | 37.42 ± 0.96 | 26.97 ± 1.19 |
| Teacher fine-tuning | **43.32 ± 0.83** | 31.29 ± 2.19 | 35.17 ± 2.68 | 36.58 ± 0.97 | 38.16 ± 0.87 | 26.58 ± 1.85 |

architecture. We report the average student validation accuracy, trained on Gaussian noise, across five different random seeds. For the **student model** evaluation we present our results on the MNIST and CIFAR10 datasets in Figure 6. Our results show that across both datasets, the student achieves the highest accuracy for all transfer setups when only the last fully connected layer, corresponding to 0.5% of the student's parameters, is fine-tuned. This finding is aligned with the previous ablation's finding and suggests that the stronger alignment between teachers and students with minimal student adaptations yields highest performance benefits. Under the same setting, we also fine-tune various fractions of last parameters in all **teacher models** in addition to the student model. Due to the high computational overhead of fine-tuning hundreds of models, we limit our experiments to CIFAR10. We rely on Gaussian noise for the knowledge transfer between teachers and students. When fine-tuning only the student, we assume last-layer fine-tuning of the teachers and vice versa when fine-tuning only the teachers. Our results in Table 3 show that fine-tuning only the last layer in both teachers and student, achieves the best result. We hypothesize that these findings are mainly due to the fact that the knowledge transfer relies on random OOD data. When initialized with the same weights, different models still exhibit a consistent behavior on such random data, as we show in Table 2. This is beneficial both for having more consistent predictions between the teachers, and a better transfer from teachers to students. However, the more the models are fine-tuned, the more they diverge in terms of their behavior on the random data, making private knowledge distillation more difficult.

**Using Current Statistics During Inference is Crucial.** The second crucial part of DIET-PATE is the use of the current statistics during inference. When using the pretrained models with the original statistics, then the gain over PATE is only marginal, as shown by the blue line in Figure 4. Only the use of the **current statistics** allow for successful knowledge transfer due to the mitigation of the covariate shift in the teachers as seen by the red and orange line. Using the current statistics also further aligns the teachers when queried on the same batch of transfer data, as all share the same mean and standard deviation over that batch. These results confirm, that DIET-PATE's performance relies on the synergy between joint initialization and the use of the current statistics.

### 5.4 IMPROVED EFFICIENCY, PRIVACY, AND UTILITY WITH DIET-CAPC

**Efficient Inference.** We demonstrate that our standard inference on unencrypted data in DIET-CaPC is orders of magnitude faster than private inference on encrypted data in CaPC (see Table 13). In these experiments, we use the same small CryptoNet-ReLU model used in CaPC (with two convolutional layers), along with ResNet10 (also used in CaPC) and ResNet18, to evaluate inference speed. Even for significantly larger models than those used in CaPC, such as ResNet18, our approach achieves much faster inference by avoiding the computational overhead of private inference. By removing

Table 4: **Performance of DIET-CaPC against CaPC.** We set $\varepsilon = 6$, $\delta = 10^{-5}$ for MNIST and $\varepsilon = 10$, $\delta = 10^{-5}$ for both CIFAR10 and TissueMNIST. For DIET-CaPC, we pretrain all MNIST and TissueMNIST teachers on StyleGAN, and the CIFAR10 teachers on Shaders21k. The models in CaPC are trained from scratch, following (Choquette-Choo et al., 2021). We report test accuracy in CaPC over the improved teachers and in DIET-CaPC for the new student (%).

| Setup | MNIST | CIFAR10 | TissueMNIST |
|---|---|---|---|
| DP-FL | 85.7% ± 0.22% | 18.5% ± 0.22% | 48.25% ± 0.12% |
| CaPC: Greedy Teacher | 94.79 ± 0.0070 | 40.11 ± 0.0124 | 53.9 ± 0.0055 |
| CaPC: Fair Teachers | 85.59 ± 0.0390 | 39.54 ± 0.0122 | 35.48 ± 0.0340 |
| DIET-CaPC | 89.83 ± 0.0010 | 41.90 ± 0.0130 | 48.99 ± 0.0001 |

the need for encrpyted inference, DIET-CaPC removes a bottleneck of CaPC, allowing a standard forward pass on GPU. Table 13 highlights that inference drops from multiple seconds to milliseconds. DIET-CaPC supports more diverse and larger teacher or student models, thus opening PATE-based collaborative learning to more complex problems while enabling dramatically faster computation.

**Improved Privacy-Utility Trade-Offs.** We also compare our DIET-CaPC to CaPC in terms of privacy-utility trade-offs for both the **Greedy Teacher** setup, where a single teacher consumes the entire privacy budget to improve their local model, and the **Fair Teachers** setup, where the privacy budget is equally split between all teachers. Our full experimental setup is specified in Appendix A.5.

**CaPC: Greedy Teacher.** In this case, we assume that one teacher uses the entire privacy budget. This yields the maximum accuracy that can be achieved by a single party in CaPC. Note that while this scenario *can* occur in practice, as CaPC lacks a built-in fairness mechanism to track the privacy budget consumed by each teacher, it is very *unrealistic*. This is because no other party would benefit from the collaboration and, hence, there would be a lack of incentive to participate in it all together. Yet, the scenario can serve as a theoretical upper bound on utility that can be achieved. In our experiments, we give the greedy teacher all additional private query samples and let them query until they exhaust the privacy budget. We then use the labeled data from the collaboration to further fine-tune their model. We report the mean accuracy over the 10 random seeds and the standard deviation.

**CaPC: Fair Teachers.** In this more realistic setup, we equally split the private query data over the teachers. Each teacher obtains the same fraction of the privacy budget $\varepsilon$. Since the privacy budget does not linearly compose in $(\varepsilon, \delta)$-DP, we divide the budget inside Rényi-DP (Mironov, 2017). Since this is the notion that is used in the CaPC and PATE internal privacy accounting, this does not add any overhead. Each teacher can query until they reach their fraction of privacy budget in Rényi-DP. At this point, they have to stop, even when they still have unanswered queries.

Our results in Table 4 highlight that our DIET-CaPC *significantly* outperforms the fair teachers in all cases. The fact that DIET-CaPC does not match the upper bound performance of the greedy teacher for MNIST and TissueMNIST stems from the discrepancy of training data of their respective evaluated models. In the case of CaPC, the *teacher* models (whose performance is reported) have been trained on the party's initial private data and the additional new private query data **from the same distribution** (up to 9k data points for MNIST and CIFAR10, and 42k for TissueMNIST). In contrast, our DIET-CaPC *student* model has not seen any single data points from the private training distribution. Instead, it was trained purely on the labeled programatically generated data, and performs inference at test time using the activation alignment. Overall, the results highlight that in realistic collaborative setups, DIET-CaPC outperforms CaPC.

**Comparison to DP-FL.** We additionally compare DIET-CaPC to central DP-federated learning (DP-FL) (McMahan et al., 2018), where the clients clip their gradients and a coordinating server adds noise to the clipped and aggregated gradients. We compare the performance of our DIET-CaPC and DP-FL with the same privacy budget and the same number of collaborating parties/teachers. We report the DP-FL hyperparameters in Table 10. Our results in Table 4 show that DIET-CaPC outperforms DP-FL over every dataset.

# 6 CONCLUSIONS AND OUTLOOK

We introduced DIET-PATE, a novel framework that overcomes the primary limitation of the standard PATE, namely its reliance on public data from the same distribution as the private dataset to perform the knowledge transfer. The key to the success of DIET-PATE is the synergy between programmatically generated synthetic data and data-free knowledge distillation. We also show that, our distributed variant, DIET-CaPC, solves the main problems of the CaPC framework: By supporting efficient standard inference on GPUs and the training of a shared student model, DIET-CaPC facilitates faster, more performant, and scalable collaboration. Overall, our approach renders private learning feasible in sensitive distributed domains.

## ETHICS STATEMENT

As machine learning becomes prevalent in sensitive domains, protecting privacy of its underlying training data is of high significance. While the PATE and CaPC framework offer formal privacy guarantees for machine learning, their practical applicability are limited due to the unavailability of public data from the same distribution as the private data or costly private inference. Our work makes a significant contribution by alleviating these problems. We make the privacy guarantees of PATE and CaPC accessible for sensitive applications. We believe that more data owners can now benefit from rigorous protection of their data.

## REPRODUCIBILITY STATEMENT

We provide comprehensive implementation details to ensure reproducibility of our results. Information about experimental configuration, including hyperparameters, model architecture, datasets and evaluation setup are given in Appendix A. Our experiments are performed on Ubuntu 22.04, with Intel(R) Xeon(R) Gold 6330 CPU and NVIDIA A40 Graphics Card with 40 GB of memory. To run our experiments we used CUDA Version 12.5 and Python 3.12.4.

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

Figure 7: Examples of programmatically generated data.

# A EXPERIMENTAL SETUP

## A.1 DATASETS

For the generation of the synthetic datasets, we use the code by the authors of Kataoka et al. (2021); Baradad et al. (2021; 2022). We downloaded the datasets dead leaves mixed and stylegan oriented from `http://data.csail.mit.edu/noiselearning/zipped_data/large_scale/dead_leaves-mixed.zip` and `http://data.csail.mit.edu/noiselearning/zipped_data/large_scale/stylegan-oriented.zip`. For Shaders21k we used the codes previously available under `http://data.csail.mit.edu/synthetic_training/shaders21k/all_codes.zip`. For FractalDB we generated the synthetic data using 1000 categories with a weight of 0.4, fill rate of 0.2 and image size of 28 pixels.

We present the additional examples of programmatically generated data used as public data in Figure 7.

We compute the *Kernel Inception Distance* (KID) (Bińkowski et al., 2021) between the original private data and the synthetic datasets in Table 5. We use the KID scores to order the synthetic data in Figure 4 and find that the PATE performance increases with less distance between private data and synthetic data.

Table 5: **KID scores.** We compute the scores for all programmatically generated data that we utilize for the knowledge transfer.

| Dataset | StyleGAN | Dead Leaves | Shaders21k | FractalDB | Gaussian noise |
|---|---|---|---|---|---|
| MNIST | 0.196 | 0.266 | 0.271 | 0.504 | 0.53 |
| CIFAR10 | 0.149 | 0.307 | 0.085 | 0.4116 | 0.7599 |
| TissueMNIST | 0.174 | 0.2963 | 0.1526 | 0.464 | 0.62 |

Table 6: **Teacher accuracy on MNIST.** We report the average teacher performance on MNIST.

| Architecture | Number of teachers | Average Teacher Validation Accuracy |
|---|---|---|
| *ResNet10* | 200 | 85.91% |
| *ResNet18* | 200 | 89.34% |

## A.2 MODELS

In line with Standard PATE (Papernot et al., 2018a), we also experiment with ResNet10 in Table 9. Additionally, we include ResNet18. Teacher accuracies for MNIST are presented in Table 6.

For DIET-PATE, the programmatically generated datasets are used to pretrain a ResNet18 model, which serves as the starting point for training both the teachers and the student in DIET-PATE. Pretraining is conducted for either 75 epochs on StyleGAN or 15 epochs on Shaders21k using the SimCLR (Chen et al., 2020) self-supervised learning framework, with a learning rate of $3 \times 10^{-4}$, two views per image, a batch size of 256, and a feature dimension of 128.

The data is resized to match the final task dimensions before training. Based on performance, we select the Shaders21k-pretrained backbone for CIFAR10 and the StyleGAN-pretrained backbone for MNIST and TissueMNIST. Specifically, for CIFAR10, we resize the Shaders21k data to 32×32×3, while for the grayscale MNIST task, we resized the StyleGAN data to 28×28×1. After pretraining, only the last layer is fine-tuned on the sensitive private data for teachers and the transfer data + private label for the student.

## A.3 HYPERPARAMETERS

We detail all hyperparameters of our experiments in Table 7. Through extensive analysis of the privacy budget, we derived different privacy parameters for each dataset, depending on the number of teachers and the consensus of the transfer dataset. For the synthetic datasets we used the available GitHub repositories to download pre-generated data, if available, or generated them ourselves. The number of samples per synthetic dataset are detailed in Table 7b. For the synthetic pretraining we use the standard data-views from SimCLR, namely **RandomResizedCrop** to the final size, i.e. 28x28 for the pre-training for MNIST and TissueMNSIT as well as 32x32 for the pretraining of CIFAR10. Then a **RandomHorizontalFlip**, a **RandomColorJitter** with a probability of 0.8, a **GaussianBlur** with a kernel size of $0.1 * image\_size$. For CIFAR10 pretraining we also apply a **RandomGrayscale** with probability of 0.2.

Table 7: Hyperparameters used in our experiments. We use the standard Adam optimizer (Kingma & Ba, 2015).

| | Num Teachers | $T$ | $\sigma_1$ | $\sigma_2$ |
|---|---|---|---|---|
| MNIST | 200 | 150 | 120 | 40 |
| CIFAR10 | 50 | 50 | 30 | 15 |
| TissueMNIST | 250 | 170 | 100 | 40 |
| Histopathologic Cancer | 50 | 50 | 30 | 15 |
| CheXpert | 50 | 50 | 30 | 15 |

(a) PATE Hyperparameters

| | Num Gen Samples |
|---|---|
| Shaders21k | 1300000 |
| StyleGAN | 105000 |
| Dead Leaves | 105000 |

(b) Training Hyperparameters (for pre-training)

| | Epochs | LR | Optimizer | Batch Size | Pretraining Dataset |
|---|---|---|---|---|---|
| MNIST | 50 | $10^{-3}$ | Adam | 256 | StyleGAN-oriented |
| CIFAR10 | 50 | $10^{-3}$ | Adam | 128 | Shaders21k MixUp |
| TissueMNIST | 50 | $10^{-3}$ | Adam | 256 | StyleGAN-oriented |
| Histopathologic Cancer | 50 | $10^{-3}$ | Adam | 128 | Shaders21k MixUp |
| CheXpert | 50 | $10^{-3}$ | Adam | 128 | Shaders21k MixUp |

(c) Training Hyperparameters (from scratch)

Table 8: Accuracies of teachers on different pretraining backbones. We evaluate the teacher models on the test data of the target data set, after they were fine-tuned on their share of private data.

| Dataset | Number of teachers | Pretraining dataset | Avg. Teacher Accuracy |
|---|---|---|---|
| MNIST | 200 | Dead Leaves Mixed | 84.46% |
| MNIST | 200 | StyleGAN-oriented | 85.13% |
| MNIST | 200 | Shaders21k MixUp | 76.14% |
| CIFAR10 | 50 | Shaders21k | 46.72% |
| TissueMNIST | 250 | Dead Leaves Mixed | 46.5% |
| TissueMNIST | 250 | StyleGAN-oriented | 48.8% |
| TissueMNIST | 250 | Shaders21k MixUp | 43.57% |

Table 9: The resulting number of labels returned and the student accuracy for the standard knowledge transfer from PATE (no pretraining) and ResNet10 using different transfer datasets for MNIST. We set the following parameters for PATE: $T = 150$, $\sigma_1 = 120$, $\sigma_2 = 40$, $\delta = 10^{-5}$ for all datasets. $T = 200$, $\sigma_1 = 100$, $\sigma_2 = 20$, $\delta = 10^{-5}$ for FMNIST (Fashion MNIST dataset). *There is not enough public data to fulfill the whole privacy budget, $\varepsilon = 6.47$. Note that OS denotes original statistics and CS represents current statistics.

| Transfer-Dataset | $\varepsilon =5$ | $\varepsilon =8$ | $\varepsilon =10$ | $\varepsilon =20$ |
|---|---|---|---|---|
| MNIST + OS | 2882 | 4631* | - | - |
| MNIST + CS | 2927 | 4717* | - | - |
| Noise + OS | 2059 | 4514 | 6578 | 19713 |
| Noise + CS | 1222 | 2732 | 3942 | 11967 |
| FractalDB + OS | 2381 | 5291 | 7682 | 23036 |
| FractalDB + CS | 1548 | 3465 | 5023 | 15033 |
| Shaders21k + OS | 1986 | 4454 | 6350 | 19351 |
| Shaders21k + CS | 1983 | 4400 | 6393 | 19232 |
| Leaves + OS | 1389 | 3191 | 4663 | 13907 |
| Leaves + CS | 1674 | 3700 | 5416 | 16316 |
| StyleGAN + OS | 1839 | 4054 | 5889 | 17843 |
| StyleGAN + CS | 2017 | 4468 | 6521 | 19673 |
| FMNIST + OS | 724 | 1569 | 2316 | 6989 |
| FMNIST + CS | 849 | 1885 | 2745 | 8268 |

(a) Number of labels returned.

| Transfer-Dataset | $\varepsilon =5$ | $\varepsilon =8$ | $\varepsilon =10$ | $\varepsilon =20$ |
|---|---|---|---|---|
| MNIST + OS | 95.2% ± 0.3% | 95.8*% ± 0.8% | - | - |
| MNIST + CS | 95.9% ± 0.4% | 96.6*% ± 0.5% | - | - |
| Noise + OS | 9.6% ± 0.4% | 9.3% ± 0.1% | 9.4% ± 0.1% | 9.7% ± 0.4% |
| Noise + CS | 34.8% ± 2.7% | 52.6% ± 2.1% | 61.8% ± 1.0% | 76.7% ± 1.8% |
| FractalDB + OS | 10.9% ± 0.8% | 13.1% ± 1.2% | 12.7% ± 1.6% | 18.2% ± 1.0% |
| FractalDB + CS | 58.5% ± 1.7% | 72.2% ± 1.4% | 75.9% ± 1.2% | 85.3% ± 0.9% |
| Shaders21k + OS | 25.7% ± 1.6% | 27.4% ± 1.3% | 30.3% ± 1.1% | 33.0% ± 0.9% |
| Shaders21k + CS | 45.9% ± 3.0% | 55.1% ± 1.9% | 57.1% ± 2.4% | 65.1% ± 2.1% |
| Leaves + OS | 34.1% ± 2.3% | 37.9% ± 4.4% | 40.1% ± 3.7% | 44.4% ± 1.2% |
| Leaves + CS | 58.7% ± 0.8% | 68.1% ± 2.8% | 72.7% ± 2.1% | 82.7% ± 0.9% |
| SytleGAN + OS | 36.2% ± 1.3% | 36.9% ± 1.5% | 40.1% ± 1.3% | 44.6% ± 2.8% |
| SytleGAN + CS | 50.1% ± 2.6% | 58.3% ± 2.4% | 62.5% ± 2.8% | 71.1% ± 1.8% |
| FMNIST + OS | 40.0% ± 0.7% | 43.5% ± 1.0% | 45.9% ± 1.6% | 50.9% ± 1.1% |
| FMNIST + CS | 54.2% ± 1.1% | 59.8% ± 2.2% | 63.7% ± 1.3% | 75.2% ± 0.9% |

(b) Student accuracy.

## A.4 DP-FL HYPERPARAMETERS

We additionally compare DIET-CaPC to a differentially private distributed learning setting, namely central DP-federated learning (FL) (McMahan et al., 2018). In central DP-FL, a joint model is trained by a central server and sent to each collaborating party. Each party then performs local training, and obtains a gradient, clips them, and shares them with the server. The server adds noise to the gradients, before updating the global model.

We use the same number of federated parties in central DP-FL, as we have teacher parties in DIET-CaPC. The hyperparameters for each setting are reported in Table 10. We implement central DP-FL using the pfl-research library (Granqvist et al., 2024) and use the Gaussian privacy mechanism (Balle & Wang, 2018) for the privacy accounting.

Table 10: Hyperparameters used for the DP-FL evaluation.

| Dataset | Model architecture | Number of parties | $\varepsilon$ | $\delta$ | central iterations | local epochs | clipping bound | local learning rate | central learning rate | noise scale |
|---|---|---|---|---|---|---|---|---|---|---|
| MNIST | ResNet18 | 200 | 6 | 1e-5 | 50 | 2 | 1.0 | 0.01 | 1.0 | 1.0 |
| CIFAR10 | ResNet9 | 50 | 10 | 1e-5 | 50 | 2 | 1.0 | 0.01 | 0.1 | 1.0 |
| TissueMNIST | ResNet18 | 250 | 10 | 1e-6 | 50 | 2 | 1.0 | 0.01 | 1.0 | 1.0 |

## A.5 CaPC UTILITY EVALUATION SETUP

**MNIST.** For the evaluation of MNIST, we assume 200 teachers, each instantiated as a ResNet18 model. The 60k training data points are evenly split into 200 subsets of 300 samples each, serving as individual training data for the teachers. We fine-tune only the last layer of each model using these subsets. Following CaPC, we designate 9k samples from the test set as additional private data, allowing teachers to query each other, obtain labels, and refine their own models. The remaining 1k test samples are used for performance assessment. Since the order of queries in PATE and CaPC influences the number of queries that can be answered within a given privacy budget (*e.g.,* queries rejected due to low consensus incur lower costs, allowing more queries overall), we repeat our experiment across 10 different random seeds. These seeds correspond to different random orderings of private samples for the greedy teacher baseline and varying assignments of the 9k samples across the 200 teachers for the fair baseline. For distributed DIET-PATE, we discard the 9k in-distribution queries and instead use StyleGAN-generated synthetic data while maintaining the same privacy budget ($\varepsilon = 10$). This additional in-distribution data provides CaPC with a practical advantage over DIET-PATE. Distributed DIET-PATE utilizes ResNet18 models pretrained on StyleGAN.

**TissueMNIST.** TissueMNIST is evaluated on 250 teachers, instantiated as ResNet18 model. The 165k training data points are split evenly among the teachers. We assume 90% (42k samples) of the test set as private data, on which the teachers can query each other. The rest is used to evaluate the performance of the models. For distributed DIET-PATE we use ResNet18 models trained on StyleGAN and discard the 42k in-distribution samples. Instead we use StyleGAN-oriented synthetic data with the same privacy budget.

**CIFAR10.** For the evaluation of CIFAR10, we assume 50 teachers, each instantiated as a ResNet18 model. We split the 50k training datapoints evenly into 50 subsets of 1k samples each, serving as individual training data for the teachers. 9k samples of the test set are chosen as additional private data and the remaining 1k are left for performance assessment. For distributed DIET-PATE we use ResNet18 models pretrained on Shaders21k and utilize Shaders21k synthetic data for the student training instead of the in-distribution data.

# B ADDITIONAL EXPERIMENTS

## B.1 MODEL ARCHITECTURE

Table 11: We compare PATE and DIET-PATE in a setting **without** public data using different model architectures. The target dataset is MNIST and the transfer dataset is StyleGAN with a privacy budget of $\varepsilon = 6, \delta = 10^{-5}$.

| | ResNet18 | VGG9 | DenseNet121 |
|---|---|---|---|
| DIET-PATE | 89.83% | 74.42% | 36.19% |
| PATE | 25.33% | 13.73% | 15.10% |

We investigate the performance of different model architectures in Table 11 and show that while ResNet18 performs best, our results are still transferable to other model architectures like VGG9. This aligns with the findings from the data-free KD paper (Raikwar & Mishra, 2022). The performance drop comes from a difference

in teacher performance after pretraining, yet our DIET-PATE still outperforms PATE when there is no public data available. Note that our experiments with ResNet18 strike a good balance between the task and model complexity (VGG9 has a small learning capacity while we do not have enough data for DenseNet121).

## B.2  STUDENT INFERENCE ABLATION

We perform additional experiments on the batch size required to achieve the best possible performance with the student model when leveraging the current statistics. We train the DIET-PATE student as ResNet18 model using Gaussian noise as transfer data for the MNIST and CIFAR10 datasets. In Figure 8 we report the validation accuracy of the student when using the current statistics with different batch sizes. We find that already a batch size of 16 leads to stable and high performance.

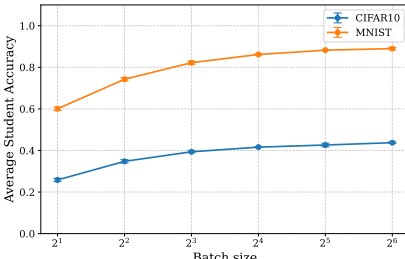

Figure 8: **To fully leverage the current statistics we only need a batch size of at least 16.** We report the student validation accuracy on the MNIST and CIFAR10 dataset when the student was trained using Gaussian Noise.

To increase utility for any single sample, a model provider can fill the mini-batch with other IID data for inference. Since inference on models is extremely fast, especially in comparison to training, this adds only negligible compute overhead.

## B.3  CHOICE OF TRANSFER DATA

We provide a full comparison between the consensus density of teachers in Figure 9. We find that initializing the teachers with the same weights increase the overall consensus, as can be seen in Figure 9a and  Figure 9b. The effect is more pronounced in the teachers with the same backbone in our DIET-PATE and DIET-CaPC framework, as can be seen in Figure 9c. In this setting, the alignment between the internal behavior of the teachers further amplifies the consensus. Higher consensus directly reduces the privacy cost per query and enables more queries to be answered, ultimately enhancing the knowledge transfer in the DIET-PATE and DIET-CaPC framework.

## B.4  COMPARISON TO DP-SGD

Table 12: We compare DP-SGD with PATE and DIET-PATE in a setting **without** public data. For the different private dataset with $\varepsilon = 6, \delta = 10^{-5}$ for MNIST and $\varepsilon = 10, \delta = 10^{-5}$ for CIFAR10 and TissueMNIST, with StyleGAN as transfer data.

|  | PATE | DIET-PATE | DP-SGD | PATE (public data) |
|---|---|---|---|---|
| MNIST | 25.33% | 89.57% | 96.85% | 96.52% |
| CIFAR10 | 17.78% | 40.56% | 42.60% | 30.36% |
| TissueMNIST | 38.53% | 48.79% | 51.64% | 52.73% |

Table 12 shows the comparison between DIET-PATE, PATE and DP-SGD in a centralized private setting with no available public data. While DP-SGD shows better performance than DIET-PATE, DIET-PATE still beats standard PATE by a large margin.

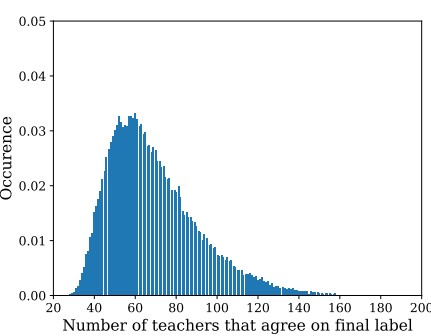

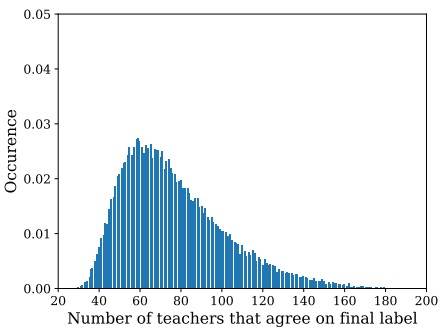

(a) Consensus density plot for teachers on Gaussian noise with different initializations of weights. $\mu = 69.3$, $\sigma = 22.19$.

(b) Consensus density plot for teachers on Gaussian noise with the same initialization of weights. $\mu = 76.88.6$, $\sigma = 25.5$

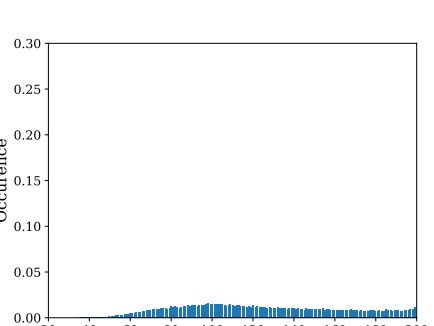

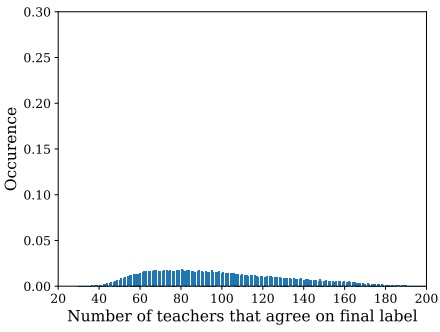

(c) Consensus density plot for DIET-PATE teachers on Gaussian noise. $\mu = 124.3, \sigma = 38.8$

(d) Consensus density plot for teachers on public FashionMNIST data with the same initialization of weights. $\mu = 98.6$, $\sigma = 32.8$

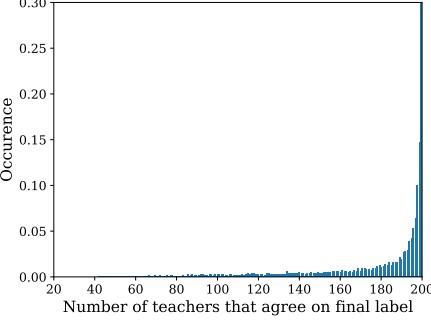

(e) Consensus density plot for teachers on public MNIST data with the same initialization of weights. $\mu = 181.29$, $\sigma = 29.4$

Figure 9: **A full comparison between the consensus density of teachers.** $\mu$ is the mean number of teachers that agree on a given label and $\sigma$ is the standard deviation of the number. We use 200 teachers and the current statistics. We observe that significantly fewer teachers with different initial weights ($\mu = 69.3$ in plot a) agree on a label compared to teachers initialized with the same weights ($\mu = 76.88$ in plot b).

## B.5 DIET-CAPC IMPROVES INFERENCE

In Table 13 we show that our DIET-CaPC, which can perform standard inference on GPU is magnitudes fast than CaPC, which has to perform costly encrypted inference on CPU. We detail the experimental setting in Appendix A.5.

Table 13: **The standard inference in DIET-CaPC is orders of magnitude faster than private inference in CaPC.** We measure the wall-clock time (sec) for private inference in CaPC vs standard inference in DIET-CaPC. We vary the modulus range, $N$, which denotes the maximum value of a given plain text number to increase the maximum security level possible in CaPC (based on its HE-transformer library (Boemer et al., 2020), which supports private inference only on CPUs). We use the CryptoNet-ReLU model provided by HE-transformer and standard ResNet10 and ResNet18 architectures.

| Method (format) | Compute | Model | Batch Size | Forward pass (sec) |
|---|---|---|---|---|
| CaPC (encrypt N=8k) | CPU | CryptoNet-ReLU | 1 | $14.22 \pm 0.11$ |
| CaPC (encrypt N=16k) | CPU | CryptoNet-ReLU | 1 | $29.46 \pm 2.34$ |
| CaPC (encrypt N=32k) | CPU | CryptoNet-ReLU | 1 | $57.26 \pm 0.39$ |
| DIET-CaPC (plain) | CPU | CryptoNet-ReLU | 1 | $0.00038 \pm 0.0006$ |
| DIET-CaPC (plain) | GPU | CryptoNet-ReLU | 1 | $0.00017 \pm 0.0008$ |
| DIET-CaPC (plain) | GPU | ResNet10 | 1 | $0.0027 \pm 0.0066$ |
| DIET-CaPC (plain) | GPU | ResNet10 | 32 | $0.0045 \pm 0.0075$ |
| DIET-CaPC (plain) | GPU | ResNet18 | 1 | $0.0041 \pm 0.0065$ |
| DIET-CaPC (plain) | GPU | ResNet18 | 32 | $0.0048 \pm 0.0049$ |

## C EXTENDED BACKGROUND

### C.1 PATE

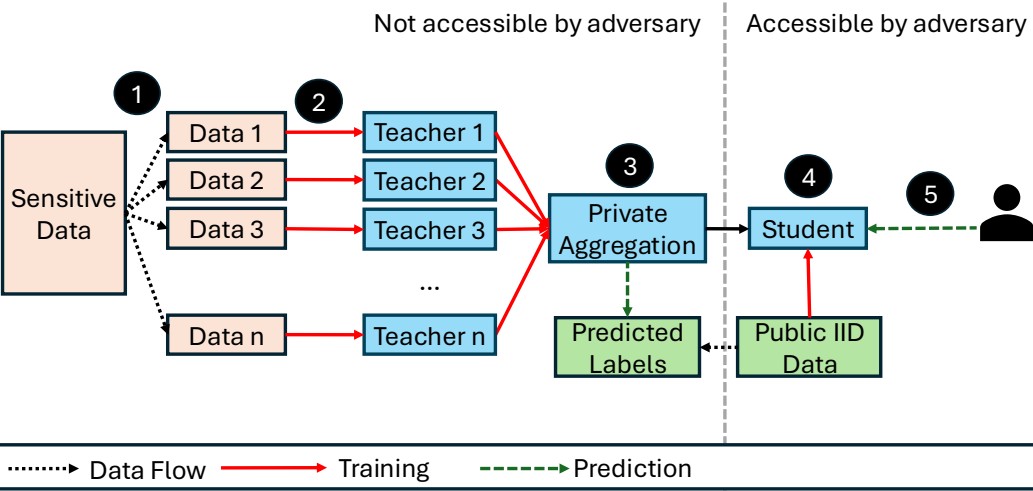

Figure 10: **Overview of the PATE framework** (taken from (Papernot et al., 2018a)). ❶ The private sensitive data is divided into partitions (in a mathematical sense, where there are no overlaps between the data partitions. ❷ An ensemble of teachers is trained on disjoint subsets of the sensitive data. ❸ The unlabeled public data from the same distribution as the sensitive data (IID) is queried against the teacher ensemble to obtain the predictions of noisy labels. ❹ A student model is trained on public data labeled using the teacher ensemble. ❺ The student model is publicly queried.

The Private Aggregation of Teacher Ensemble (PATE) (Papernot et al., 2017; 2018a) framework, visualized in Figure 10, protects the privacy of training data by transferring knowledge from an ensemble of teacher models to a student model. The privacy of the teacher training data is protected under DP guarantees.

The private training data is split into $n$ disjoint subsets, where $n$ is the number of teachers. Then every teacher is trained independently on a single subset. During inference, every teacher predicts a label. To protect the privacy of the teacher training data during prediction, a private aggregation is performed. The intuition behind the aggregation is, that when there is a strong consensus among teachers, then the label does not depend on a single teacher and thus not on a single data point. Hence the collective prediction is intuitively private. To provide DP guarantees, the aggregation mechanism adds carefully calibrated Gaussian noise to the vote count

histogram of the aggregated teacher votes and then returns the label with the highest number of votes. This noisy vote aggregation is the Gaussian NoisyMax (GNMax).

An improvement over the naive GNMax aggregation is the Confident-GNMax, which filters out queries for which teachers do not have high consensus. First the algorithm checks if the plurality vote crosses a threshold T. Privacy is enforced by adding Gaussian noise with a variance of $\sigma_1^2$, which is much higher compared to the noise added in the usual GNMax mechanism afterwards, which has variance $\sigma_2^2$. For queries that don't pass the threshold $\perp$ is returned.

The final step in PATE is the training of a student model via knowledge transfer using *unlabled* **public** data. Every answered query by the teacher ensemble increases the incurred privacy cost. After supervised training, the student can the be made public without incurring additional privacy cost on the private data, as it has only seen the public data and privacy-preserving labels.

## C.2 PRIVACY ANALYSIS FOR PATE

Here we provide the some of the theorems and propositions needed for the privacy analysis of the GNMax mechanism proposed by Papernot et al. (2018a) and used by Choquette-Choo et al. (2021), DIET-PATE and DIET-CaPC. The proofs will not be repeated here, but can be seen in (Papernot et al., 2018a).

**Teacher models:** Each teacher is trained independently on a subset of the private data. The data is partitioned such that no pair of teachers will have trained on overlapping data.

**Aggregation mechanism:** The GNMax aggregation mechanism is defined as:

$$\mathcal{M}_\sigma(x) \triangleq \mathrm{argmax}_i\{n_i(x) + \mathcal{N}(0, \sigma^2)\},$$

where for a sample $x$ and classes 1 to $m$, $f_j(x)$ denotes the $j$-th teacher model's prediction and $n_i(x)$ denotes the vote count for the $i$-th class (*i.e.*, $n_i(x) = |\{j : f_j(x) = i\}|$). Papernot et al. (2017) use a data-dependent analysis for the GNMax using Theorem 3 in conjunction with Proposition 4.

**Proposition 1.** *The GNMax aggregator $\mathcal{M}_\sigma$ guarantees $(\lambda, \lambda/\sigma^2)$-RDP for all $\lambda \geq 1$.*

**Proposition 2.** *For a GNMax aggregator $\mathcal{M}_\sigma$ the teachers' vote histogram $\bar{n} = (n_1, \ldots, n_m)$, and for any $i^* \in [m]$, we have*

$$\mathbf{Pr}[\mathcal{M}_\sigma(D) \neq i^*] \leq q(\bar{n}),$$

*where*

$$q(\bar{n}) \triangleq \frac{1}{2} \sum_{i \neq i^*} \mathrm{erfc}\left(\frac{n_{i^*} - n_i}{2\sigma}\right).$$

**Theorem 3.** *Let $\mathcal{M}$ be a randomized algorithm with $(\mu_1, \varepsilon_1)$-RDP and $(\mu_2, \varepsilon_2)$-RDP guarantees and suppose there exists a likely outcome $i^*$ given a dataset $D$ and a bound $\tilde{q} \leq 1$ such that $\tilde{q} \geq \mathbf{Pr}[\mathcal{M}_\sigma(D) \neq i^*]$. Additionally suppose that $\lambda \leq \mu_1$ and $\tilde{q} \leq \exp(\mu_2 - 1)\varepsilon_2 / \left(\frac{\mu_1}{\mu_1 - 1} \cdot \frac{\mu_2}{\mu_2 - 1}\right)^{\mu_2}$. Then, for any neighboring dataset $D'$ of $D$ we have:*

$$D_\lambda(\mathcal{M}(D)||\mathcal{M}(D')) \leq \frac{1}{\lambda - 1} \log\left((1 - \tilde{q}) \cdot \boldsymbol{A}(\tilde{q}, \mu_2, \varepsilon_2)^{\lambda - 1} + \tilde{q} \cdot \boldsymbol{B}(\tilde{q}, \mu_1, \varepsilon_1)^{\lambda - 1}\right),$$

*where $\boldsymbol{A}(\tilde{q}, \mu_2, \varepsilon_2) \triangleq (1 - \tilde{q})/\left(1 - (\tilde{q}e^{\varepsilon_2})^{\frac{\mu_2 - 1}{\mu_2}}\right)$ and $\boldsymbol{B}(\tilde{q}, \mu_1, \varepsilon_1) \triangleq e^{\varepsilon_1}/\tilde{q}^{\frac{1}{\mu_1 - 1}}$.*

**Proposition 4.** *For any $i^* \in [m]$, we have $\mathbf{Pr}[\mathcal{M}_\sigma(D) \neq i^*] \leq \frac{1}{2} \sum_{i \neq i^*} \mathrm{erfc}(\frac{n_{i^*} - n_i}{2\sigma})$, where $\mathrm{erfc}$ is the complementary error function.*

## C.3 CAPC

The Confidential and Private Collaborative (CaPC) learning (Choquette-Choo et al., 2021) provides DP-guarantees for federated learning in a collaborative setting. We present the CaPC protocol, following (Choquette-Choo et al., 2021) in Figure 11. It leverages multi-pary computation (MPC) (Yao, 1986), homomorphic encryption (HE) and other techniques in combination with privately aggregated teacher models, with a privacy accounting following the PATE framework.

**Homomorphic encryption.** HE defines an encryption scheme such that he encryption and decryption functions are homomorphic between plaintext and cyphertext spaces, i.e. all computations can be performed on either the plain- or cyphertext and have the same outcome.

**Secret sharing.** Secret sharing denotes a scheme where a datum, the secret is divided into parts across different parties, such that each party only has one part of the secret. The secret can only be recovered if a certain number of parties combine their shares.

**Multi-party computation.** Secure MPC has the goal to create a method, such that parties can jointly compute a function over their inputs, while keeping those inputs private. This concept is most important in CaPC, as we want to ensure the privacy of each unlabeled private sample. The specialized MPC in CaPC is used to securely evaluate a private ML model on private data. HE and Yao's garbled-circut protocol is used for confidential two-party deep learning inference. Then noise is added to provide DP gurantees, and finally the querying party and the privacy guardian run Yao's garbled circuit to obtain the argmax of querying party.

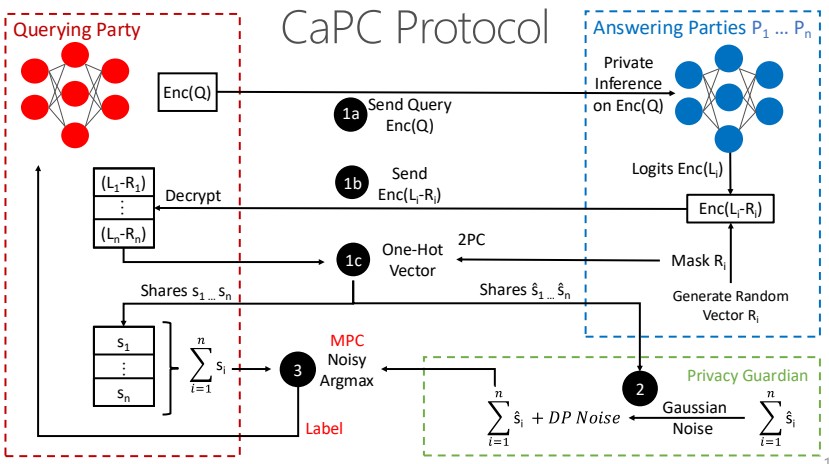

Figure 11: **Overview of the CaPC protocol** (taken from (Choquette-Choo et al., 2021)). **1a** Querying party $\mathcal{P}_{i_*}$ sends encrypted query $Q$ to each answering party $\mathcal{P}_i$, $i \neq i_*$. Each $\mathcal{P}_i$ engages in a secure 2-party computation protocol to evaluate $\mathsf{Enc}(Q)$ on the model $\mathcal{M}_i$ and outputs encrypted logits $\mathsf{Enc}(\boldsymbol{L}_i)$. **1b** Each answering party, $\mathcal{P}_i$, generates a random vector $\boldsymbol{R}_i$, and sends $\mathsf{Enc}(\boldsymbol{L}_i - \boldsymbol{R}_i)$ to the querying party, $\mathcal{P}_{i_*}$, who decrypts to get $\boldsymbol{L}_i - \boldsymbol{R}_i$. **1c** Each answering party $\mathcal{P}_i$ runs Yao's garbled circuit protocol ($Y_i$) with querying party $\mathcal{P}_{i_*}$ to get $\boldsymbol{s}_i$ for $\mathcal{P}_{i_*}$ and $\hat{\boldsymbol{s}}_i$ for $\mathcal{P}_i$ s.t. $\boldsymbol{s}_i + \hat{\boldsymbol{s}}_i$ is the one-hot encoding of argmax of logits. **2** Each answering party sends $\hat{\boldsymbol{s}}_i$ to the privacy guardian. The privacy guardian sums $\hat{\boldsymbol{s}}_i$ from each $\mathcal{P}_i$ and adds Laplacian or Gaussian noise for DP. The querying party sums $\boldsymbol{s}_i$ from each $Y_i$ computation. **3** The privacy guardian and the querying party run Yao's garbled circuit $Y_s$ to obtain argmax of querying party and PG's noisy share. The label is output to the querying party.

## C.4 DATA-FREE KNOWLEDGE TRANSFER

Using data from a different distribution as input data results in a shift of the activations in a neural network. This shift was defined as the covariate shift. To tackle the problem, Raikwar & Mishra (2022), uses the batch norm layers and mitigates the covariate shift by normalizing the current batch according to the mean and standard deviation of the current batch.

Consider the case of a BatchNorm layer placed after a hidden layer $\mathcal{H}$ within a neural network trained on the original dataset $D$. Let $B$ represent a batch of images sampled from $D$, $G$ a batch of random samples from a Gaussian distribution, $\boldsymbol{h}$ be the vector of activations for some neuron $n$ belonging to $\mathcal{H}$, $m$ be the batch size and $P(\boldsymbol{h})$ be the activation distribution of that $n^{th}$ neuron over the batch. Due to the covariate shift, Raikwar & Mishra (2022) observe that: $P(\boldsymbol{h}|G) \neq P(\boldsymbol{h}|B)$. Generally in inference the original statistics $(\mu_r, \sigma_r)$ are used to calculate the normalized activations $\bar{\boldsymbol{h}}$ and the learned parameters $(\gamma, \beta)$ to calculate the scaled activations $\hat{\boldsymbol{h}}$. This leads to:

$$P(\bar{\boldsymbol{h}}|G, \mu_r, \sigma_r) \neq P(\bar{\boldsymbol{h}}|G, \mu_r, \sigma_r)$$
$$\text{Thus, } P(\hat{\boldsymbol{h}}|G, \mu_r, \sigma_r, \gamma, \beta) \neq P(\hat{\boldsymbol{h}}|G, \mu_r, \sigma_r, \gamma, \beta)$$

However if the current batch statistics $(\mu_B, \sigma_B)$ are used for the normalization step, then this aligns the two different distributions, such that even though $P(\boldsymbol{h}|G) \neq P(\boldsymbol{h}|B)$ holds true:

$$P(\bar{\boldsymbol{h}}|G, \mu_B, \sigma_r) = P(\bar{\boldsymbol{h}}|G, \mu_B, \sigma_B)$$
$$\approx \mathcal{N}(0, 1)$$

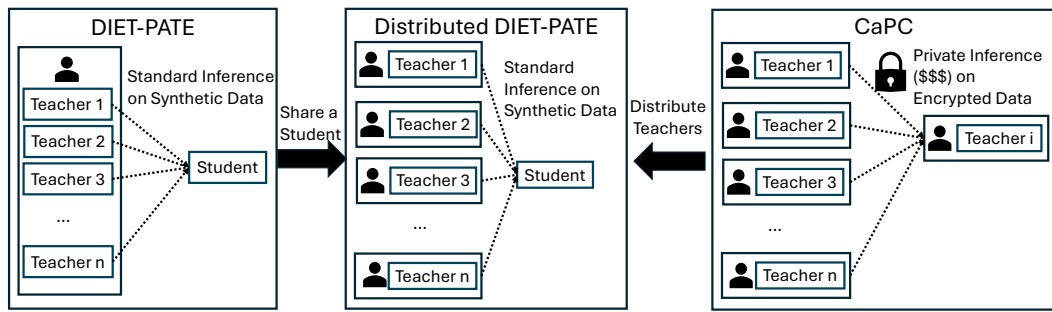

Figure 12: **DIET-CaPC** combines the strengths of both, DIET-PATE and CaPC frameworks. The distributed DIET-PATE follows the approach from CaPC, where *teachers are distributed across many parties*, enabling collaborative learning. While CaPC allows only to improve an individual model for a participating party, DIET-PATE follows the standard PATE, where all the teachers collaborate to train a *shared student* model, thus providing more benefits to all collaborators with a higher overall performance gain. DIET-PATE also solves the other main issue in CaPC, where the teachers perform very costly *private inference on encrypted private data*. Instead, we leverage the non-private synthetic data that are programmatically generated, which allows us to perform efficient standard inference.

$$\text{Hence, } P(\hat{\bm{h}}|G, \mu_B, \sigma_B, \gamma, \beta) = P(\hat{\bm{h}}|G, \mu_B, \sigma_B, \gamma, \beta)$$

This approach reduces the covariate shift due to a different input distribution from the expected one at every BatchNorm layer in the network.

## D    DETAILS ON DIET-CAPC

**Overview.** We present the overview of DIET-CaPC and how it combines the strengths of both, DIET-PATE and CaPC frameworks in Figure 12.

**Protocol.** We present the protocol for DIET-CaPC from the CaPC perspective (Figure 11). In distributed DIET-PATE, in step **1a** of our protocol, an unlabeled programmatically generated sample $x$ is sent to all collaborating (answering) parties, which in this case act like teachers in PATE; subsequently each of the teachers runs standard inference on the sample $x$. The output logits $\bm{L}_i$ from the models $\mathcal{M}_i$ in all the answering parties are unencrypted (returned in the plain form). We compute the one-hot-encoding $\bm{V}_i$ of the logits $\bm{L}_i$. **1b** The answering party generates secret shares of the one-hot encoding $\bm{V}_i$, distributing $\bm{s}_i$ to the student party and $\hat{\bm{s}}_i$ to the privacy guardian. **2** The privacy guardian sums $\hat{\bm{s}}_i$ and adds Gaussian noise for DP. The student party sums $\bm{s}_i$ from each teacher party $\mathcal{P}_i$. **3** The privacy guardian and the student party run Yao's garbled circuit to obtain argmax of added shares from the student party and the noisy shares from the privacy guardian. The label is returned to the student party, which trains the student model on all the privately labeled data samples $x$-s.

**Threat Model.** Our DIET-CaPC relies on the same assumptions and threat model as the original CaPC (Choquette-Choo et al., 2021). We require a semi-trusted party called the privacy guardian, which does not collude with any party. When one or more parties are corrupted, this has no impact on the confidentiality guarantee of the CaPC and DIET-CaPC protocol, but the privacy budget $\varepsilon$ will degrade faster because the sensitivity of the aggregation mechanism increases. We also note that standard deployments of CaPC are used in sensitive use-cases with participants, like hospitals or banks, that have a genuine interest in training high-utility models to solve their tasks. Therefore, similar to CaPC, we do not consider and protect against the risk of poisoned model updates (Biggio et al., 2012) intended to worsen the model quality.

## E    LLM USAGE DECLARATION

We used large language models to refine some texts in our manuscript, focusing exclusively on style, grammar, and spelling, without compromising the original semantic meaning.

