# OpenReview forum: "Distributed PATE and CaPC on a DIET: Private Knowledge Transfer without Public Data or Private Inference"
_ICLR.cc/2026/Conference — Submitted to ICLR 2026_

### Official Review · Reviewer_6ZA9 · 2025-10-27

**Soundness:** 3
**Presentation:** 2
**Contribution:** 2
**Rating:** 4
**Confidence:** 4

**Summary:**

This paper proposes DIET-PATE, which enables PATE-style private knowledge transfer without relying on public in-distribution data by leveraging programmatically generated data and data-free knowledge distillation. It further extends the approach to a distributed setting, replacing expensive encrypted inference with standard inference and enabling a shared student model.

**Strengths:**

1. The paper addresses a practical limitation of PATE by removing the dependence on public in-distribution data. The extension to distributed learning improves efficiency by eliminating costly encrypted inference.
2. The paper is clearly written with a well-structured methodology description, making the technical workflow easy to follow.

**Weaknesses:**

1. The method stays close to the classic teacher-student paradigm for privacy. It mainly combines known techniques rather than introducing a clear new algorithmic idea.
2. The efficiency analysis does not account for the computational and engineering cost of generating the synthetic data at scale.
3. While the introduction mentions finance and other sensitive domains, the experiments use very basic benchmarks like MNIST (TissueMNIST, while biomedical, is still a standardized image-classification benchmark). This weakens claims about generalization to real-world settings.

**Questions:**

Have the authors evaluated or estimated the computational and engineering cost of generating synthetic data at scale, and how would this overhead impact the claimed efficiency gains in practical deployments?

Can the authors demonstrate the practicality of their approach on more realistic and domain-relevant datasets to support the generalization claims?

---

> ### Author Response · Authors · 2025-11-20
> **Rebuttal to Reviewer 6ZA9**
>
> We thank the reviewer for their constructive feedback. We appreciate that the reviewer finds our paper to be well written.
>
> In summary we show the *scalability* of DIET-PATE and evaluate the cost of generating the synthetic data at scale.
>
> We address the points in detail below:
>
> >**W1: It mainly combines known techniques rather than introducing a clear new algorithmic idea.**
>
> The novelty of our framework lies in making PATE and its distributed variant DIET-CaPC fully independent of the need for in-distribution public data. PATE’s reliance on public data has, since its introduction, been considered as PATE’s biggest practical limitation. Our contribution overcomes this limitation, and thereby, makes these frameworks practical.
>
> For the design of our approach, we had to identify the synergy between programmatically generated data and data-free knowledge distillation, supported by (1) joint teacher-student initialization, increasing the ensemble consensus and (2) the use of the current batch statistic. Identifying this design itself is a significant contribution.
>
>
>
>
>
> >**W2:  The efficiency analysis does not account for the computational and engineering cost of generating the synthetic data at scale.**
>
>
> We timed the programmatical generation for 1,000 such images of the synthetic datasets and report the required time here. Note that the generation can also be parallelized.
>
> | Dataset | Dead Leaves | StyleGAN | FractalDB | Gaussian noise |
> |---|---|---|---|---|
> | Time  | 6min 01s | 1min 17s | 2min 15s | 0.02s |
>
> The generation is a one-time operation and the samples can be reused across many tasks in the future. Additionally all of the synthetic datasets are openly available and can simply be downloaded with the links provided in the paper or in the github repositories of the respective works. The results highlight that there is negligible generation overhead.
>
>
> >**W3: While the introduction mentions finance and other sensitive domains, the experiments use very basic benchmarks like MNIST (TissueMNIST, while biomedical, is still a standardized image-classification benchmark). This weakens claims about generalization to real-world settings.**
>
> We performed additional experiments using the TinyViT model (20.6 million parameters) and a histopathologic cancer dataset [1], with images of size 96x96x3. We trained 50 teacher models for PATE and 50 teacher models for DIET-PATE, where the latter are pretrained using Shaders21k.
>
> We report the average accuracy of our DIET-PATE student against the PATE student in % with different transfer datasets across three random seeds. We used a privacy budget of $(10, 5 \times 10^{-6})$-DP. We report the full hyperparameters in Appendix A.3.
>
>
> | Method | Gaussian noise | Shaders21k | Public data |
> |---|---|---|---|
> | DIET-PATE | $81.68 \pm 0.39$ | $81.37 \pm 0.53$ | $82.54 \pm 0.18$ |
> | PATE | $49.48 \pm 5.04$ | $64.97 \pm 1.26$ | $76.40 \pm 1.10$ |
>
> The results highlight our approach’s ability to scale both in terms of data and model complexity.
>
> We would greatly appreciate updating the rating if the above responses address Reviewer's concerns.
>
>
> **References**
>
> [1] Will Cukierski. Histopathologic Cancer Detection. https://kaggle.com/competitions/histopathologic-cancer-detection, 2018. Kaggle.

---

> > ### Author Response · Authors · 2025-11-27
> > **Rebuttal Follow Up**
> >
> > We kindly follow up regarding the rebuttal and check whether our responses address the Reviewer’s concerns.
> >
> > In particular, we have:
> > - scaled DIET-PATE to more complex tasks and a larger model architecture, namely a histopathologic cancer dataset (96$\times$96$\times$3) with TinyViT. Additionally in response to Reviewer kUUg we added CheXpert (320$\times$320$\times$3). Our results show that DIET-PATE performs much better than PATE, highlighting its applicability in complex settings where public data is usually not available;
> > - shown the small cost for generating synthetic samples, and highlighted that we did not have to generate them because they can be directly downloaded from the internet. This together enables a practical use of that data for DIET-PATE.
> >
> > We remain happy to discuss any questions or clarify any points further. We thank the Reviewer again for their time and valuable feedback.

---

### Official Review · Reviewer_kUUg · 2025-10-29

**Soundness:** 3
**Presentation:** 2
**Contribution:** 2
**Rating:** 2
**Confidence:** 4

**Summary:**

The paper introduces DIET-PATE, which removes the dependence of PATE on in-distribution public data by combining programmatically generated synthetic data and data-free knowledge distillation. Teacher and student models are initialized on synthetic data, teachers fine-tune on private subsets, and knowledge transfer is performed on synthetic samples using the current batch normalization statistics to mitigate distribution shift. The paper also proposes DIET-CaPC, which integrates DIET-PATE into the collaborative CaPC framework to replace encrypted private inference with efficient plaintext synthetic inference, thereby enabling joint student training across parties.

**Strengths:**

- The work addresses the limitation of PATE, which is a dependence on public data, therefore broadening applicability to domains like healthcare.
- A good engineering contribution by extending the setting into a collaborative setting.

**Weaknesses:**

- In Sec 3, the paper states that the current BN statistics must always be used during inference, but it's unclear how this works for single-sample inference, where BN statistics may be unstable.
- The transition from teacher-student to collaborating parties is confusing to the reader. The mapping between entities is unclear and should be introduced more gradually and consistently.
- The method's success heavily depends on the quality of the generated data. If the synthetic data distribution is too far from private data, transfer may fail; this limitation is not clearly discussed.
- Despite claims of enabling learning in complex, sensitive domains, experiments use simple datasets (MNIST, CIFAR-10, TissueMNIST). These are insufficient and unconvincing to validate the claims.
- TissueMNIST results underperform DP-SGD and lack a discussion of practical relevance for medical tasks.
- Table 2 is redundant, and it is not necessary to include it; the gap is expected, and a textual summary would suffice.
- One other important point is about DIET-CaPC. While integrating it into a collaborative learning (CL) algorithm is a natural extension, the paper treats CaPC as the only CL framework. This is a narrow view, as a broad landscape of CL paradigms exists, which could equally benefit from the proposed mechanism. A stronger paper would evaluate or at least discuss why CaPC was chosen over other frameworks, and how DIET-PATE could integrate with or outperform more scalable, widely used methods. As written, the work doesn't convincingly argue its generality.

**Questions:**

Please refer to the above weaknesses.

---

> ### Author Response · Authors · 2025-11-20
> **Rebuttal 1 to Reviewer kUUg**
>
> We thank the reviewer for their valuable and insightful feedback.
>
> In summary we show the *scalability* of DIET-PATE and evaluate our DIET-CaPC against a distributed baseline.
>
> We address all points below:
>
>
> >**W1: In Sec 3, the paper states that the current BN statistics must always be used during inference, but it's unclear how this works for single-sample inference, where BN statistics may be unstable.**
>
>
> We separate our answer to this question to the two parts in our framework, the **teacher parties** and the **student party**.
>
> 1. The teacher parties are used to label the data points for the student with the DP-aggregation mechanism. Here we want to label **as-many datapoints as possible** while staying within the ($\varepsilon$, $\delta$)-privacy budget. In this setting it is both unreasonable and unpractical to use only a single-sample for inference. Especially when using Gaussian noise for knowledge transfer, there is no limit on the amount of data that can be used and thus no limit on the batch size.
> 2. The student party is used to perform inference on new data, while protecting the privacy of the training data. It targets domains, where multiple queries are naturally processed, e.g. analyzing multiple patient records. The goal of our framework is that the student model can be used over multiple queries.
>
> We additionally performed an ablation study on the required batch size needed for a good validation accuracy for the student model. The student model was trained using Gaussian noise and the labels from the teachers. During inference we use the current statistics to mitigate the covariate shift. We found that a batch size of 16 is sufficient for good validation accuracy, reported below in %.
>
> |  | 2 | 4 | 8 | 16 | 32 | 64 |
> |---|---|---|---|---|---|---|
> | MNIST | 60.04 $\pm$ 0.87 | 74.24 $\pm$ 0.77 | 82.21 $\pm$ 0.77 | 86.17 $\pm$ 0.32 | 88.23 $\pm$ 0.18 | 88.98 $\pm$ 0.57 |
> | CIFAR10 | 25.82 $\pm$ 0.81 | 34.76 $\pm$ 0.81 | 39.36 $\pm$ 0.48 | 41.6 $\pm$ 0.18 | 42.62 $\pm$ 0.87 | 43.74 $\pm$ 0.41 |
>
>
> These results and additional discussion have been added to Appendix B.2.
>
>
>
> >**W2: The transition from teacher-student to collaborating parties is confusing to the reader. The mapping between entities is unclear and should be introduced more gradually and consistently.**
>
> The collaborating parties in CaPC are equivalent to the teachers in PATE, with the difference that PATE operates in a **centralized** setting, whereas CaPC operates in a **distributed** setting. Since CaPC is distributed and operates on sensitive data it was impossible for the original protocol to share this data to train a unified student model.
> Our DIET-CaPC *removes this limitation* and enables to train a shared student model, similar to PATE
> We added a more detailed description of the mapping to Sections 4.1 and 4.2.
>
>
>
>
> >**W3: The method's success heavily depends on the quality of the generated data. If the synthetic data distribution is too far from private data, transfer may fail; this limitation is not clearly discussed.**
>
> The success of our method does **not** depend on the quality of generated data, as we show throughout the experimental Section 5, especially in Figure 4 and Figure 5 in the original submission. Our DIET-PATE achieves nearly the same accuracy with Gaussian noise (worst case OOD data), as with public in distribution data.
>
> Additionally the synthetic data that we use and display in Figure 6 in the Appendix are from a distinct distribution as the private data. This is also shown by the blue and green lines in Figure 4, where the original PATE framework can not transfer sufficient knowledge to the student. Hence, the success of our framework does not depend on the quality of the programmatically generated data, in fact, it works well even when using Gaussian noise.
>
> >**W4: Despite claims of enabling learning in complex, sensitive domains, experiments use simple datasets (MNIST, CIFAR-10, TissueMNIST). These are insufficient and unconvincing to validate the claims.**
>
> We performed additional experiments using the TinyViT model (20.6 million parameters) and a sensitive medical dataset, namely a histopathologic cancer dataset [1], with images of size 96x96x3. We trained 50 teacher models for PATE and 50 teacher models for DIET-PATE, where the latter are pretrained using Shaders21k.
>
> We report the average accuracy of our DIET-PATE student against the PATE student in % with different transfer datasets across three random seeds. We used a privacy budget of $(10, 5 \times 10^{-6})$-DP. We report the full hyperparameters in Appendix A.3.
>
>
> | Method | Gaussian noise | Shaders21k | Public data |
> |---|---|---|---|
> | DIET-PATE | $81.68 \pm 0.39$ | $81.37 \pm 0.53$ | $82.54 \pm 0.18$ |
> | PATE | $49.48 \pm 5.04$ | $64.97 \pm 1.26$ | $76.40 \pm 1.10$ |
>
>
> The results, added to the new Section 5.2 in the main paper, highlight our approach’s ability to scale both in terms of data and model complexity.

---

> > ### Author Response · Authors · 2025-11-20
> > **Rebuttal 2 to Reviewer kUUg**
> >
> > >**W5: TissueMNIST results underperform DP-SGD and lack a discussion of practical relevance for medical tasks.**
> >
> > The results on DP-SGD were performed on a central setting using the available training data to train the model. In distributed settings, DP-SGD is not directly applicable. In practice, hospitals rarely have enough training data for machine learning models, since the labeling needs to be done by medical experts and is hence, very expensive. For practical use-cases, distributed frameworks, like our DIET-CaPC are more relevant. We show the performance of DP-SGD-based distributed learning in our answer to W7 below and highlight that DIET-CaPC outperforms that distributed variant of DP-SGD.
> >
> >
> > >**W6: Table 2 is redundant, and it is not necessary to include it; the gap is expected, and a textual summary would suffice.**
> >
> > We moved Table 2 to the Appendix and extend the textual summary of the **Efficient Inference** paragraph in the corresponding Section.
> >
> >
> >
> > >**W7: A stronger paper would evaluate or at least discuss why CaPC was chosen over other frameworks, and how DIET-PATE could integrate with or outperform more scalable, widely used methods. As written, the work doesn't convincingly argue its generality.**
> >
> > Our framework solves two limitations in CaPC, namely (1) costly encrypted inference and (2) dependence on public in-distribution data. We show that our solution does not result in a performance degradation, but rather holistically improves over the CaPC method.
> >
> > To further address the reviewer’s comment, we performed additional experiments to compare our DIET-CaPC against other DP-based distributed learning. Concretely, we evaluated against central DP federated learning (DP-FL) [2]. In central DP-FL, a joint model is trained by a central server and sent to each collaborating party. Each party then performs local training, and obtains a gradient, clips them, and shares them with the server. The server adds noise to the gradients, before updating the global model.
> >
> > We compare the performance of the final students model in our DIET-CaPC and the trained shared model in DP-FL with the same privacy budget and the same number of collaborating parties/teachers. We report the full hyperparameters in Appendix B.4.
> >
> >
> > |  | MNIST ($6, 10^{-5}$)-DP | CIFAR10 ($10, 10^{-5}$)-DP | TissueMNIST ($10, 10^{-5}$)-DP |
> > |---|---|---|---|
> > | DP-FL | 85.70\% | 18.50\% | 48.25\% |
> > | DIET-CaPC | 89.93\% | 41.90\% | 48.99\% |
> >
> >
> > The results, added to Section B.4, highlight that our distributed DIET-CaPC outperforms DP-FL over every dataset.
> >
> >
> >
> >
> > We would greatly appreciate updating the rating if the above responses address Reviewer's concerns.
> >
> > **References**
> >
> > [1] Will Cukierski. Histopathologic Cancer Detection. https://kaggle.com/competitions/histopathologic-cancer-detection, 2018. Kaggle.
> >
> > [2] H. Brendan McMahan, Daniel Ramage, Kunal Talwar, and Li Zhang. Learning differentially private recurrent language models. In International Conference on Learning Representations, 2018.

---

> ### Comment · Reviewer_kUUg · 2025-11-24
>
> I would like to thank the authors for their rebuttal and clarifications. While some points have been resolved by the authors, I remain unconvinced by certain responses and have the following questions/doubts:
>
> > e.g., analyzing multiple patient records
>
> I remain unconvinced by the response to this concern.
>
> What if the hospital does not possess multiple patient records at the given moment? What if they have a new patient, and it's the only one at the moment? Should they wait for new customers to come in, and then proceed with the inference? Or should they use other patient details (to fill the batch) unnecessarily, thereby using more computing power, just to make an inference for one patient?
>
> In many cases, this is both realistic and common. A single new patient may appear at any time. The model must work robustly even for a single query.
>
> - the authors either need to report this as a weakness of the paper, or
> - provide a good use case for this.
>
> > [W3, W4]
>
> For this W3 and W4, please provide experiments with high-resolution images; the ones included in the paper and the rebuttal are still unconvincing. Options could include chestx-ray, ham10000 datasets with high resolutions, since these are the real datasets used in practice.
>
> > W7
>
> Regarding this, my point was conceptual: why is DIET-PATE applied exclusively to CaPC when many other collaborative learning frameworks exist? Was it because CaPC is an extension of PATE? Can you generalize it to other methods?
>
> I look forward to your response on these.
>
> Best,
> Reviewer kUUg

---

> > ### Author Response · Authors · 2025-11-26
> > **Response to Reviewer kUUg**
> >
> > We thank the Reviewer for further engaging in the rebuttal and are happy to address their further questions and comments.
> >
> > >**What if the hospital does not possess multiple patient records at the given moment? Or should they use other patient details (to fill the batch) unnecessarily, thereby using more computing power, just to make an inference for one patient?**
> >
> > To increase utility for any single patient, the hospital can fill the mini-batch with other patients’ data for inference.
> >
> > To address the Reviewer’s concern about the use of compute power for this practice, we report the number of Giga FLOPs needed for a forward pass on the large TinyViT model at different batch sizes.
> >
> >
> > | Batch Size | 1 | 8 | 16 | 32 | 64 |
> > |:---:|:---:|:---:|:---:|:---:|:---:|
> > | Giga FLOPs  | 8.759 | 70.072 | 140.144 | 280.289 | 560.578 |
> >
> >
> > Given that, according to the specification [1], a modern-day GPU (NVIDIA A40, as used for the experiments), peaks at 34.7 **Terra** FLOPs, i.e, 34,700.00 Giga FLOPs per second, even with a batch size of 64, the required run time is 0.016 seconds, hence adding minimal compute overhead.
> > We added a discussion about this to Appendix B.2.
> >
> > [1] https://images.nvidia.com/content/Solutions/data-center/a40/nvidia-a40-datasheet.pdf
> >
> > >**Please provide experiments with high-resolution images; the ones included in the paper and the rebuttal are still unconvincing.**
> >
> > To address the Reviewer’s comment, we performed additional experiments on the CheXpert dataset., which consists of 224,316 images with a 320$\times$320$\times$3 resolution. Since DIET-PATE is not designed for multi-label classification, we instead perform a 3-class (negative, positive, unclear) classification for the *Lung Opacity* label. Therefore, we train 50 TinyViT teachers on 80% of the CheXpert dataset, where the DIET-PATE teachers are pre-trained using Shaders21k and only the final layer is fine-tuned. We use a privacy budget of $(10, 5\times 10^{-6})$-DP to transfer knowledge to the student model and report the performance in % based on three different transfer datasets.
> >
> > |  | Gaussian noise | Shaders21k | Public data |
> > |:---:|:---:|:---:|:---:|
> > | DIET-PATE | 68.5916 $\pm$ 2.4585 | 68.7120 $\pm$ 0.8720  | 70.8446 $\pm$ 0.6959 |
> > | PATE | 44.3285 $\pm$ 2.9526 | 50.5469 $\pm$ 3.3020 | 48.9886 $\pm$ 3.6135 |
> >
> >
> > The results highlight that our DIET-PATE can transfer knowledge about high-resolution medical-images with OOD data such as Gaussian noise or Shaders21k, achieving nearly the same accuracy as with the IID data (Public Data). This highlights that our DIET-PATE fully removes the requirement of public data from the PATE framework, also in challenging domains.
> >
> > >**why is DIET-PATE applied exclusively to CaPC when many other collaborative learning frameworks exist? Was it because CaPC is an extension of PATE?**
> >
> > DIET-PATE is applied to CaPC because CaPC is the distributed equivalent of PATE. CaPC performs collaborative learning based on sharing private labels. Other collaborative learning frameworks, such as DP-FL, exchange model gradients instead. Our additional results from the rebuttal, added to Appendix B.4, highlight that our DIET-CaPC outperforms DP-FL.

---

> > > ### Author Response · Authors · 2025-11-27
> > > **Rebuttal Follow Up**
> > >
> > > We kindly follow up regarding the rebuttal and check whether our responses address the Reviewer’s concerns.
> > >
> > > In particular, we have:
> > > - scaled DIET-PATE to more complex tasks and a larger model architecture, namely a histopathologic cancer dataset (96$\times$96$\times$3) with TinyViT and CheXpert (320$\times$320$\times$3). Our results show that DIET-PATE performs much better than PATE, highlighting its applicability in complex settings where public data is usually not available;
> > > - added experiments to quantify that our method has good performance with small mini-batches and that using such mini-batches has negligible overhead of 0.016 seconds;
> > > - discussed the application of the DIET-PATE framework to CaPC.
> > >
> > > We remain happy to discuss any questions or clarify any points further. We thank the Reviewer again for their time and valuable feedback.

---

### Official Review · Reviewer_mmbT · 2025-11-01

**Soundness:** 3
**Presentation:** 3
**Contribution:** 3
**Rating:** 6
**Confidence:** 2

**Summary:**

This paper proposes DIET-PATE, a framework designed to address the limitation of the PATE framework's reliance on in-distribution public data. It achieves private knowledge transfer by leveraging programmatically generated data and data-free knowledge distillation. Furthermore, this approach is extended to distributed collaborative learning with CaPC, eliminating the need for costly private inference on encrypted data. Experimental results demonstrate DIET-PATE's effectiveness.

**Strengths:**

1. The paper is well-structured, presenting a clear delineation of the proposed methodology, experimental setup, and results.
2. The proposed DIET-CaPC circumvents the prohibitive cost of private inference in CaPC by replacing encrypted private queries with programmatically generated data. This design is novel and well-executed, enhancing both engineering feasibility and practical applicability.

**Weaknesses:**

1. In Section 4.2 (DIET-CaPC), the paper claims that "the shared student model can then also be used to label new locally obtained data and improve teacher i, without incurring any additional privacy cost." I'm not really doubtful of the reasonableness of the claims, but I think the paper should quantify and qualify them better.

2. How should the relationship between DIET-CaPC and the original CaPC be understood? It appears to prioritize training efficiency and model shareability over personalized learning for querying parties. Should this be viewed as a shift in objectives rather than a technical extension?

**Questions:**

Please refer to Weaknesses.

---

> ### Author Response · Authors · 2025-11-20
> **Rebuttal to Reviewer mmbT**
>
> We thank the reviewer for the positive and constructive feedback. We appreciate that our DIET-CaPC is considered *well-executed* and that the reviewer finds our paper to be well structured.
>
> We address the points in detail below:
>
> >**W1: The paper claims that "the shared student model can then also be used to label new locally obtained data and improve teacher i, without incurring any additional privacy cost." I'm not really doubtful of the reasonableness of the claims, but I think the paper should quantify and qualify them better.**
>
> As the labels that the student relies on for training are obtained with differential privacy, the student model benefits from the so-called **postprocessing guarantees**, a well-established concept that formulates that any data-independent transformation applied to the output of a differentially private mechanism preserves the **original privacy guarantee**. Therefore, neither the training of the student model, nor its use to label additional data infers any additional privacy costs.
>
> The reason why we use the student to label additional data is that the student has some non-trivial accuracy on the task. Hence, it can label new locally obtained data without the need of expensive human-expert labeling, which will result in more training data for the local teacher, improving its performance.
>
> >**W2: How should the relationship between DIET-CaPC and the original CaPC be understood? It appears to prioritize training efficiency and model shareability over personalized learning for querying parties. Should this be viewed as a shift in objectives rather than a technical extension?**
>
> The overall objective between DIET-CaPC and the original CaPC remains the same: distributed collaborating parties want to get access to a strong machine learning model to solve their downstream tasks.
>
> DIET-CaPC improves upon how to achieve such a model over the original CaPC. Instead of improving each collaborator’s local model separately, which results in suboptimal utility, as we highlighted in Table 3 of the original submission, it first trains a powerful *joint* student model. Given again the differential privacy postprocessing guarantees, each teacher could optionally copy the student model at the end of training and fine-tune it further, for example, using the DP-SGD algorithm, on their own local data for better personalization.
>
>
> We would greatly appreciate updating the rating if the above responses address Reviewer's concerns.

---

> > ### Comment · Reviewer_mmbT · 2025-11-26
> >
> > Thanks for the response. I keep my first-round score.

---

> > > ### Author Response · Authors · 2025-11-26
> > > **Answer to Reviewer mmbT**
> > >
> > > We thank the Reviewer for considering our rebuttal and maintaining their positive rating. Shall further questions arise, we are happy to answer.

---

### Official Review · Reviewer_kPqL · 2025-11-09

**Soundness:** 3
**Presentation:** 3
**Contribution:** 2
**Rating:** 6
**Confidence:** 2

**Summary:**

The paper proposes DIET-PATE, a variant of PATE that removes the need for public data drawn from the same distribution as the private dataset. The key idea is to (1) pretrain both teachers and student on large procedurally generated image datasets, and (2) apply data-free knowledge distillation with adapted batch-normalization statistics to reduce the covariate shift between synthetic and private data. The authors extend the approach to a distributed collaborative setting, DIET-CaPC, which integrates DIET-PATE into the CaPC framework and replaces expensive private inference on encrypted data with standard inference on synthetic queries, enabling the training of a single shared student. Experiments on MNIST, CIFAR-10, and TissueMNIST show that DIET-PATE approaches or exceeds standard PATE under comparable privacy budgets when in-distribution public data is absent, and that DIET-CaPC yields substantial efficiency gains and better privacy–utility trade-offs than CaPC.

**Strengths:**

1. Addresses a real limitation of PATE and CaPC.
The paper tackles a well-known practical bottleneck of PATE: the dependence on public in-distribution data, which is often unrealistic in sensitive domains such as healthcare or finance. Likewise, DIET-CaPC directly targets CaPC’s main pain points—slow private inference and fragmented privacy budgets leading to only modest local improvements. The problem formulation and motivation are clear and compelling.

2. Extension to distributed collaboration:
DIET-CaPC is a meaningful extension, e.g., replacing encrypted private queries with synthetic queries removes the need for HE/MPC-based private inference and allows larger models compared to the CryptoNet-style networks used in CaPC. The protocol description (teachers, student party, privacy guardian) is clear, and the empirical comparison shows both speedups and more favorable privacy–utility trade-offs.

3. Extensive evaluation and ablations: The experimental section is fairly thorough.

4. Clarity and reproducibility:
The paper is overall well-written and easy to follow. Implementation details are described in the appendix with specificity to help reproduce the experiments.

**Weaknesses:**

1. Incremental methodological novelty: It seems like DIET-PATE is an integration of PATE, programmatically generated pretraining, and data-free KD with BN “current statistics.” Each component is taken from prior work; the main novelty lies in combining them to remove PATE’s reliance on public in-distribution data and in porting this combination into CaPC. While this integration is practically useful, the methodological contribution may be seen as somewhat incremental for a top-tier venue, especially since there is no new privacy analysis, theoretical insight, or substantially new algorithmic mechanism beyond the pipeline design.

2. Limited experimental scope and scalability:
The evaluation focuses on image classification tasks with relatively standard-sized models (ResNet10/18) and modest-resolution data (MNIST, CIFAR-10, TissueMNIST). It remains unclear how DIET-PATE behaves with larger architectures or more complex modalities (e.g., high-resolution images) where pretraining and synthetic generation might be more challenging. Given that the paper emphasizes improved scalability for DIET-CaPC, additional experiments or a more in-depth discussion of scaling bottlenecks would be valuable.  ￼

3. Lack of security/robustness analysis:
The work is framed around privacy, but does not investigate robustness to adversarial manipulation of the synthetic data or the protocol. For example, what happens if the synthetic generation process is biased or partially corrupted, or if an adversary contributes malicious teachers in DIET-CaPC? Even a discussion of such risks and possible mitigations would enhance the paper’s practical relevance for sensitive domains.

4. Intuitive design choices:
In DIET-PATE, teachers fine-tune only the last layer on private data, and the student also fine-tunes a small fraction of parameters; the paper shows empirically that this helps alignment and performance, but these choices are not deeply analyzed.

**Questions:**

Please refer to 'Weaknesses'.

---

> ### Author Response · Authors · 2025-11-20
> **Rebuttal 1 to Reviewer kPqL**
>
> We greatly appreciate your valuable feedback and insightful comments, which helped us further improve our submission.
>
> In summary, we extended the experimental setting and show the *scalability* of DIET-PATE in terms of data and model complexity, we clarified the *threat model* of DIET-CaPC and gave additional insights into *fine-tuning only the last layer*.
>
> We address all points in detail below:
>
> >**W1:  The methodological contribution may be seen as somewhat incremental for a top-tier venue, especially since there is no new privacy analysis, theoretical insight, or substantially new algorithmic mechanism beyond the pipeline design.**
>
> The novelty of our framework lies in making PATE and its distributed variant DIET-CaPC fully independent of the need for in-distribution public data. PATE’s reliance on public data has, since its introduction, been considered as PATE’s biggest practical limitation. Our contribution overcomes this limitation, and thereby, makes these frameworks practical.
>
> For the design of our approach, we had to identify the synergy between programmatically generated data and data-free knowledge distillation, supported by (1) joint teacher-student initialization, increasing the ensemble consensus and (2) the use of the current batch statistic. Identifying this design itself is a significant contribution.
>
> >**W2: It remains unclear how DIET-PATE behaves with larger architectures or more complex modalities**
>
> We performed additional experiments using the TinyViT model (20.6 million parameters) and a histopathologic cancer dataset [1], with images of size 96x96x3. We trained 50 teacher models for PATE and 50 teacher models for DIET-PATE, where the latter were pretrained using Shaders21k.
>
> We report the average accuracy of our DIET-PATE student against the PATE student in % with different transfer datasets across three random seeds. We used a privacy budget of $(10, 5 \times 10^{-6})$-DP and added the full hyperparameters in Appendix A.3.
>
>
> | Method | Gaussian noise | Shaders21k | Public data |
> |---|---|---|---|
> | DIET-PATE | $81.68 \pm 0.39$ | $81.37 \pm 0.53$ | $82.54 \pm 0.18$ |
> | PATE | $49.48 \pm 5.04$ | $64.97 \pm 1.26$ | $76.40 \pm 1.10$ |
>
>
>
> Our results, added to the new Section 5.2 in the main paper and displayed here, highlight our approach’s ability to scale both in terms of data and model complexity.
>
>
> >**W3: Lack of security/robustness analysis**
>
> Our work, in alignment with the original CaPC, considers honest-but-curious participants that try to learn as much information as possible but do not deviate from the protocol. This is a realistic threat model as CaPC is usually deployed in sensitive use-cases with participants, like hospitals or banks, that have a genuine interest in training high-utility models to solve their tasks and ensure, for example, their patients’ well being.
>
> Therefore, similar to CaPC, our work focuses on the  privacy protection of the collaborating parties’ data. To ensure this, we follow the CaPC protocol, which we build our DIET-CaPC on, and require a semi-trusted party called the privacy guardian. We work under the assumption that the privacy guardian does not collude with any party.
>
> In the case that one or more parties are corrupted, this has no impact on the confidentiality guarantee of the CaPC protocol, but the privacy budget $\varepsilon$ will degrade faster because the sensitivity of the aggregation mechanism increases [2].
>
> We added this discussion and the threat model to Appendix D.

---

> > ### Author Response · Authors · 2025-11-20
> > **Rebuttal 2 to Reviewer kPqL**
> >
> > >**W4: In DIET-PATE, teachers fine-tune only the last layer on private data, and the student also fine-tunes a small fraction of parameters**
> >
> > We ran additional experiments and fine-tuned various fractions of parameters for both teachers and students. As this experiment required multiple retraining of all teachers, we focused solely on the CIFAR10 dataset and the ResNet18 model.  We used Gaussian noise for the transfer from teachers to students and got the following results across 5 random seeds. We always fine-tuned from the last layer backward, e.g. fine-tuning 20% of parameters means the final 20% of parameters before the output. This ensures that the models behave as similar as possible, while allowing for more adaptation to the data.
> >
> > We report test accuracy (%) for fine-tuning either the **teachers**, the **student** or **teachers and student** with various fractions of parameters. When fine-tuning only the student, we assume last-layer fine-tuning of the teachers and vice versa when fine-tuning only the teachers.
> >
> > | Finetuned Parameters (\%) | 0.5 | 20 | 40 | 60 | 80 | 100 |
> > |---|---|---|---|---|---|---|
> > | Student fine-tuning | 43.32 $\pm$ 0.83 | 32.58 $\pm$ 0.66 | 39.61 $\pm$ 1.81 | 39.43 $\pm$ 0.77 | 40.54 $\pm$ 1.55 | 35.04 $\pm$ 2.31 |
> > | Student and Teacher fine-tuning | 43.32 $\pm$ 0.83 | 25.58 $\pm$ 1.37 | 32.45 $\pm$ 0.37 | 35.93 $\pm$ 2.83 | 37.42 $\pm$ 0.96 | 26.97 $\pm$ 1.19 |
> > | Teacher fine-tuning | 43.32 $\pm$ 0.83 | 31.29 $\pm$ 2.19 | 35.17 $\pm$ 2.68  | 36.58 $\pm$  0.97 | 38.16 $\pm$ 0.87 | 26.58 $\pm$ 1.85 |
> >
> >
> > Our results confirm that fine-tuning only the last layer (0.5% of the parameters) of the models achieves the best student model performance. We attribute this effect to the alignment between the teacher models and the student, which we analyze in Section 5.2 of the original paper. If the teacher models and the student behave more similarly, then the teacher labels for the Gaussian noise carry more information about the underlying data distribution that can help the student to create gradients that align better with the final task. Additionally higher teacher alignment leads to more answered queries, where each query requires a lower privacy budget, as we also show in Figure 9.
> >
> >
> >
> > We would greatly appreciate updating the rating if the above responses address Reviewer's concerns.
> >
> >
> > **References**
> >
> > [1] Will Cukierski. Histopathologic Cancer Detection. https://kaggle.com/competitions/histopathologic-cancer-detection, 2018. Kaggle.
> >
> > [2] Christopher A. Choquette-Choo, Natalie Dullerud, Adam Dziedzic, Yunxiang Zhang, Somesh Jha, Nicolas Papernot, and Xiao Wang. CaPC Learning: Confidential and private collaborative learning. In ICLR (International Conference on Learning Representations), 2021.

---

> > > ### Author Response · Authors · 2025-11-27
> > > **Rebuttal Follow Up**
> > >
> > > We kindly follow up regarding the rebuttal and check whether our responses address the Reviewer’s concerns.
> > >
> > > In particular, we have:
> > > - scaled DIET-PATE to more complex tasks and a larger model architecture, namely a histopathologic cancer dataset (96$\times$96$\times$3) with TinyViT. Additionally in response to Reviewer kUUg we added CheXpert (320$\times$320$\times$3). Our results show that DIET-PATE performs much better than PATE, highlighting its applicability in complex settings where public data is usually not available;
> > > - added a discussion on the security and robustness of DIET-CaPC;
> > > - included new experiments showing that fine-tuning only the last layer in the teacher and student models achieves the best performance, justifying our design choice.
> > >
> > > We remain happy to discuss any questions or clarify any points further. We thank the Reviewer again for their time and valuable feedback.

---

### Author Response · Authors · 2025-11-20
**General Response**

We would like to thank all the reviewers for their valuable feedback and insightful comments, which greatly helped us further improve our submission. We thank the reviewers for recognizing that our new DIET-PATE addresses a “real limitation” (Reviewer kPqL, kUUg, 6ZA9), solving a long standing problem of the dependence on public in-distribution data for PATE. Additionally our “meaningful extension” (Reviewer kPqL) to DIET-CaPC removes the costly encrypted inference, enhancing the practical applicability (Reviewer kPqL, mmbT, 6ZA9).  Our “extensive evaluation and ablation” (Reviewer kPqL), show that DIET-PATE approaches or exceeds standard PATE under the same privacy budgets when in-distribution public data is absent.

In the following, we present the highlights of the rebuttal.

>**DIET-PATE is effective for larger model architectures and complex medical datasets.**

We conducted additional experiments on (1) a histopathologic cancer dataset which consists of 220,000 images of size (96$\times$96$\times$3) and (2) CheXpert which consists of 224,316 images of size (320$\times$320$\times$3), with a large vision transformer (TinyViT, 20.6 million parameters). Our results, added to the new Section 5.2 and displayed below, show that our DIET-PATE generalizes to different model architectures and larger scale datasets. DIET-PATE consistently outperforms standard PATE by up to 30% (Gaussian noise, histopathologic cancer). We added all hyperparameters to Appendix A.3.


| | | **Histopathologic Cancer** | | | **CheXpert** | |
|:---:|:---:|:---:|:---:|:---:|:---:|:---:|
| **Method** | **Gaussian Noise** | **Shaders21k** | **Public data** | **Gaussian Noise** | **Shaders21k** | **Public data** |
| DIET-PATE | 81.68$\pm$0.39 | 81.37$\pm$0.53 | 82.54$\pm$0.18 | 68.59 $\pm$2.46 | 68.71$\pm$0.87 | 70.844$\pm$0.69 |
| PATE | 49.48$\pm$5.04 | 64.97$\pm$1.26 | 76.40$\pm$1.10 | 44.33$\pm$2.95 | 50.55$\pm$3.30 | 48.99$\pm$3.61 |

>**Only fine-tuning the last layer of both teachers and students yields best performance.**

We added another ablation on fine-tuning different fractions of last parameters in both the teachers and student models. As this experiment required multiple retraining of all teachers, we focused solely on CIFAR10 for ResNet18. We used Gaussian noise for the transfer from teachers to the student and always fine-tuned from the last layer backward.

We report test accuracy for fine-tuning either the **teachers**, the **student** or **teachers and student** with various fractions of parameters.

| Finetuned Parameters (\%) | 0.5 | 20 | 40 | 60 | 80 | 100 |
|---|---|---|---|---|---|---|
| Student fine-tuning | 43.32 $\pm$0.83 | 32.58$\pm$0.66 | 39.61$\pm$1.81 | 39.43$\pm$0.77 | 40.54$\pm$1.55 | 35.04$\pm$2.31 |
| Student and teacher fine-tuning | 43.32$\pm$0.83 | 25.58$\pm$1.37 | 32.45$\pm$0.37 | 35.93$\pm$2.83 | 37.42$\pm$0.96 | 26.97$\pm$1.19 |
| Teacher fine-tuning | 43.32$\pm$0.83 | 31.29$\pm$2.19 | 35.17$\pm$2.68  | 36.58$\pm$0.97 | 38.1$\pm$0.87 | 26.58$\pm$1.85 |


The results show that only when we fine-tune the last layer in teacher and student we achieve the best performance. We hypothesize that these findings are mainly due to the fact that the knowledge transfer relies on random OOD data. When initialized with the same weights, different models still exhibit more consistent behavior on such random data. This is beneficial both for having more consistent predictions between the teachers (lower incurred privacy costs) and a better transfer from teachers to students (better final utility).


>**DIET-CaPC outperforms other forms of private collaborative learning.**

We performed additional experiments comparing DIET-CaPC with central DP-federated learning (DP-FL) [1] where the clients clip their gradients and a coordinating server adds noise to the clipped and aggregated gradients.

We compare the performance  (%) of our DIET-CaPC and DP-FL with the same privacy budget and the same number of collaborating parties/teachers. We report the full hyperparameters in Appendix A.4.

| | MNIST (6, $10^{-5}$)-DP | CIFAR10 (10, $10^{-5}$)-DP | TissueMNIST (10, $10^{-5}$)-DP |
|---|---|---|---|
| DP-FL | 85.7 $\pm$ 0.22 | 18.5 $\pm$ 0.22 | 48.25 $\pm$ 0.12 |
| DIET-CaPC | 89.83$\pm$ 0.0010 | 41.90$\pm$0.0130 | 48.99$\pm$0.0001  |

We observe that our DIET-CaPC outperforms DP-FL over each tested dataset, highlighting its practical relevance for distributed learning in sensitive domains. The results were added to Section 5.4.

Finally, we clarified individual questions in an answer to the respective reviewer.
We believe that these additions fully address all the points raised by the reviewers by showing that the method is applicable to highly complex medical data and large models and outperforms the baselines both on the central and distributed learning setups.

**References**

[1] H. Brendan McMahan, Daniel Ramage, Kunal Talwar, and Li Zhang. Learning differentially private recurrent language models. ICLR, 2018.

---

### Meta-Review · Area_Chair_eb5j · 2025-12-29

**Summary:**

The paper proposes DIET-PATE, a method for removing the reliance of the PATE method on in-distribution public data. Reviewers agreed this is a valuable direction to explore, but were nonetheless critical of several aspects of the work. The authors provided comprehensive rebuttals, however, the discussion did not seem to converge toward a sufficiently strong agreement toward acceptance. The paper is therefore recommended to be revised and resubmitted.

**Reviewer Concerns:**

Reviewers were concerned about several aspects of the work. The concerns that were voiced by multiple reviewers and not completely addressed by the rebuttal were the novelty of the method and the scale of the experiments (even in light of the experiments added during the rebuttal phase).

**Reviewer Scores:**

Some reviewer discussion did manage to take place in the narrow window that was possible this year. The positive-leaning reviewers (albeit with weak accepts) attested less confidence and one stated they would not alter their score. The negative-leaning reviewers attested more confidence and while rebuttal discussions were useful, one stated that some concerns remained outstanding, and it is unlikely both would have upgraded to an accepting score. There is also a matter the volume of new experiments that can be considered acceptable in the rebuttal phase and when it has crossed the verge to necessitating a full resubmission.

---

### Decision · Program_Chairs · 2026-01-26

Reject